# Exploring and Leveraging Class Vectors for Classifier Editing

**Jaeik Kim**[1]    **Jaeyoung Do**[† 1,2,]

AIDAS Laboratory, [1]IPAI & [2]ECE, Seoul National University

† indicates corresponding author

{jake630, jaeyoung.do}@snu.ac.kr

## Abstract

Image classifiers play a critical role in detecting diseases in medical imaging and identifying anomalies in manufacturing processes. However, their predefined behaviors after extensive training make post hoc model editing difficult, especially when it comes to forgetting specific classes or adapting to distribution shifts. Existing classifier editing methods either focus narrowly on correcting errors or incur extensive retraining costs, creating a bottleneck for flexible editing. Moreover, such editing has seen limited investigation in image classification. To overcome these challenges, we introduce Class Vectors, which capture class-specific representation adjustments during fine-tuning. Whereas task vectors encode task-level changes in weight space, Class Vectors disentangle each class's adaptation in the latent space. We show that Class Vectors capture each class's semantic shift and that classifier editing can be achieved either by steering latent features along these vectors or by mapping them into weight space to update the decision boundaries. We also demonstrate that the inherent linearity and orthogonality of Class Vectors support efficient, flexible, and high-level concept editing via simple class arithmetic. Finally, we validate their utility in applications such as unlearning, environmental adaptation, adversarial defense, and adversarial trigger optimization.

## 1 Introduction

Classifiers have long been fundamental in Computer Vision (CV), applied in diverse fields from medical imaging [35] to anomaly detection [84]. With the rise of Vision Transformers (ViTs) [13, 44], their classification capabilities have significantly improved, leading to the widespread availability of fully fine-tuned models across various tasks. As a result, open platforms such as HuggingFace now offer extensive collections of classifiers, enabling plug-and-play usage for diverse applications [66]. However, even within the same task, users may have specific requirements for certain deterministic rules. For example, in disease diagnosis, some users may prioritize minimizing errors for specific conditions they handle, as even minor misclassifications can have serious consequences. Alternatively, others may require a classifier that performs reliably in their own distributional context, such as a snowy environment. Thus, a *one-size-fits-all* classifier is impractical for meeting diverse user needs within the current model supply chain. This highlights the importance of classifier editing, which modifies class-specific knowledge post hoc while preserving unrelated prior knowledge [72].

Despite its importance, classifier editing remains challenging because deeply optimized models encode rigid behaviors shaped by their training distributions. For example, data scarcity for vehicles in snowy scenes often teaches the model to adopt the shortcut *vehicle + snow → snowplow*, causing it to mislabel buses in snow as snowplows [60, 26]. Moreover, efficiently modifying classifiers with minimal data in real-world scenarios remains an open problem. Recent classifiers (*e.g.*, ViTs), for instance, require significantly more training compared to traditional CNNs [64, 68],

making knowledge modifications with few samples more difficult [39, 36] and increasing the risk of introducing new biases [67, 5]. As a result, existing classifier editing methods are computationally intensive [80] and often rely on auxiliary information such as object mask, requiring representations to be modified one by one for each image [60]. These challenges, amplified by the sparse information density of visual data [23], require defining "*where-to-edit*" and remains underexplored in vision models, whose scope is largely restricted to correcting image-wise misclassifications [62, 80].

To address these challenges, we revisit recent image classifiers through the lens of a model's adaptation to specific classes during training by introducing the novel concept of *Class Vectors*. Class Vectors capture per-class representation shifts during fine-tuning by computing the difference between the centroid representations of pretrained and fine-tuned models. Inspired by *task vectors* [29], which represent weight updates for tasks during fine-tuning, Class Vectors aim to disentangle class-specific behavior from task-wide adaptations. Although task vectors are effective for task-level applications such as model alignment [21, 6, 43, 25] and detoxification [29, 83], they inherently capture task-level modifications, limiting their applicability for fine-grained classifier editing. In contrast, Class Vectors operate at the class level and can be applied either by directly steering latent representations via a training-free approach or by mapping them back into weight space for model editing. This class-level modification alleviates existing editing constraints by replacing predictive rules across an entire class rather than adjusting the model on an image-by-image basis.

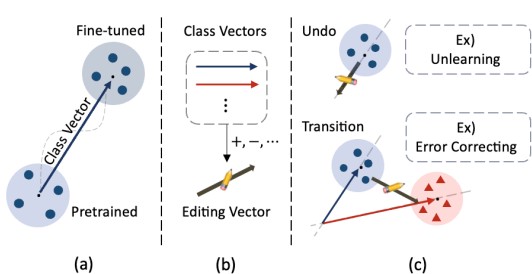

Figure 1: Class Vector and its applications. (a) Class Vector captures centroid representation adaptation in the latent space. (b) Editing vector with high-level concepts using arithmetic operations on Class Vectors. (c) Editing vectors can undo predictive behaviors by reversing the adaptation direction, or transition the classifier logic to correct errors.

Our findings reveal that linear trajectories exist along which the model adapts to specific classes during fine-tuning, forming the basis of Class Vectors. This behavior remains barrier-free despite the complexity of high-dimensional representation learning and the nonlinear characteristics of modern classifiers [57, 56], supported by *Cross-Task Linearity* (CTL) [87]. We then explore two key properties of Class Vectors that enable their effective use in classifier editing: (1) linearity and (2) independence. During inter-class interpolation, predictions and logits transition smoothly along linear paths. Furthermore, we demonstrate that modifying a target class's representations does not influence other classes, confirming that Class Vectors act independently. This behavior is supported by the *Neural Collapse* [55]: during fine-tuning, class-specific feature shifts become quasi-orthogonal, enabling targeted adjustments with minimal interference. These fundamental properties of Class Vectors enable precise editing of class-specific predictive rules through simple arithmetic operations such as addition, subtraction, and scaling (Fig. 1), offering several advantages as follows:

- **High efficiency**: Enables edits without retraining via latent steering, or for specific tasks can be trained in under 1.5 seconds using fewer than 1.5K parameters and a single sample.
- **High-level interaction**: Facilitates intuitive high-level concept editing, also allowing non-expert users to perform edits without neural network expertise.
- **Flexibility**: Provides precise control over the degree and nature of edits by adjusting the scaling coefficients of Class Vectors to align with user intentions.

We present extensive experiments demonstrating the effectiveness of Class Vectors in real-world applications such as model unlearning, adapting to unfamiliar environments, preventing typographic attacks, and optimizing triggers for backdoor attacks.

## 2  Related Work

**Adaptation vectors.** Empirical studies of neural network loss landscapes show that fine-tuning proceeds along convex, well-aligned directions in weight space [18, 73]. Building on this, *task vectors* [29], the weight delta from a pretrained model to its fine-tuned version, have been used in classifiers [76, 78], large language models [77, 81], and LoRA adapters [74, 10], enabling multi-task

learning [27, 79], detoxification [61], and style mixing [74]. Meanwhile, in large language models (LLMs), *in-context vectors* steer model outputs at inference time by encoding task- or example-level instructions as additive offsets in latent space (*e.g.*, for controllable text generation [43, 25, 47]). Such adaptation vectors typically perform model editing at the global, task-wide level. In contrast, our *Class Vectors* isolate adaptations at the per-class level in latent feature space, providing localized, class-specific control with minimal interference to other classes. Unlike task vectors, which impose global weight shifts, Class Vectors capture intrinsic, persistent representation shifts that can be mapped to weights. They differ also from in-context vectors—transient, label-agnostic offsets—and from concept activation vectors (CAVs) [33], which describe rather than edit concepts.

**Characterizing Neural Networks.** Early work [18] demonstrated that the loss landscape along the straight-line path from random initialization to a fully trained model is nearly convex, suggesting that training could follow a linear trajectory. Linear Mode Connectivity (LMC) [28, 15, 49] then showed that independently trained models on the same task maintain almost constant loss under linear weight interpolation, and the concept of task vectors [29, 53] revealed that scaling these vectors yields semantically meaningful performance changes. Layerwise Linear Feature Connectivity (LLFC) [86] extended this phenomenon to the feature space, proving that at every layer the feature maps of an interpolated model align proportionally with the linear blend of the feature maps of the original models that show LMC. More recently, Cross-Task Linearity (CTL) [87] found that models fine-tuned on different tasks still exhibit approximate linear behavior in their features under weight interpolation, and Neural Collapse (NC) described how penultimate-layer features converge to equidistant class prototypes [55]. In this work, we show that pretrained-to-fine-tuned model pairs also satisfy a CTL with feature alignment, and we use NC to establish the independence of Class Vectors.

# 3 Foundations for Class Vectors

Given $n$ data $\{x_1, x_2, \ldots, x_n\} \subset \mathcal{X}_{\text{task}}$ with corresponding $k$ labels $\{c_1, c_2, \ldots, c_k\} \subset \mathcal{Y}$, image classifier is defined as $\mathcal{M}(\cdot, \theta \in \mathbb{R}^d) : \mathcal{X}_{\text{task}} \mapsto \mathcal{Y}$, comprising an encoder $f(\cdot, \theta^{\text{e}} \in \mathbb{R}^{d_e}) : \mathcal{X}_{\text{task}} \mapsto \mathcal{Z}$ and a classification head $g(\cdot, \theta^{\text{h}} \in \mathbb{R}^{d_h}) : \mathcal{Z} \mapsto \mathcal{Y}$. Here, classifier is represented as $\mathcal{M} = g \circ f$, where $\circ$ denotes function composition. Let $\theta^e_{\text{pre}} \in \mathbb{R}^{d_e}$ represent the pretrained weight and $\theta^e_{\text{ft}} \in \mathbb{R}^{d_e}$ be the fine-tuned weight of the classifier encoder for a specific task. We first define the Class Vector.

**Definition 3.1** (Class Vector). *For a class $c$, let $S = \{s_1, \ldots, s_{|S|}\} \subset \mathcal{S}$ denote the set of its samples. The* Class Vector *$\kappa_c \in \mathbb{R}^m$ is the difference between the expected last-layer representations of the fine-tuned and the pretrained encoders (i.e., penultimate layer of models):*

$$\kappa_c = \mathbb{E}_{s \in S}\big[f\big(s, \theta^{\text{e}}_{\text{ft}}\big)\big] - \mathbb{E}_{s \in S}\big[f\big(s, \theta^{\text{e}}_{\text{pre}}\big)\big].$$

The centroid representation of a fine-tuned classifier $z^c_{\text{ft}}$ for a class $c$ can be formulated as $z^c_{\text{ft}} = z^c_{\text{pre}} + \kappa_c$, where $z^c_{\text{pre}}$ denotes pretrained centroid representation and $S = \mathcal{X}_{\text{task}}$.

## 3.1 Formal Justification

We aim to demonstrate that the class-specific changes induced by fine-tuning are captured by a *single latent vector* $\kappa_c := z^c_{\text{ft}} - z^c_{\text{pre}}$, so that merely scaling $\kappa_c$ interpolates a smooth path of class-$c$ behavior. To justify this claim, we build on two well-documented phenomena.

**(i) Task-level weight linearity.** Prior work [29] shows that the *task vector* $\tau = \theta^e_{\text{ft}} - \theta^e_{\text{pre}}$ captures a linearly meaningful direction in *weight space*: moving the weights along $\tau$, $f\big(x; \theta^e_{\text{pre}} + \lambda\tau\big)$, causes predictable performance shifts as $\lambda$ varies [18].

**(ii) Cross-Task Linearity (CTL)** [87]. When two fine-tuned checkpoints $\theta_i, \theta_j$ originate from the same $\theta_{\text{pre}}$, weight interpolation is almost equivalent to latent interpolation for *every* input $x$:

$$f(x; \alpha\theta_i + (1 - \alpha)\theta_j) \approx \alpha f(x; \theta_i) + (1 - \alpha) f(x; \theta_j).$$

CTL thus bridges weight-space linearity to latent-space linearity. In Theorem 3.1, beyond any pair of fine-tuned weights, we show that it is *even tighter* on the segment connecting the pretrained model to its fine-tuned checkpoint, and we confirm that this pretrain-to-finetune interpolation traces a semantically meaningful path in latent space.

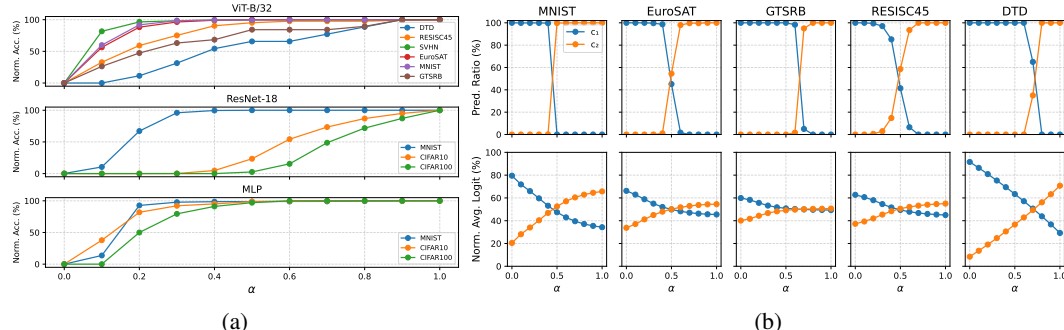

(a)                      (b)

Figure 2: (a) Line-search between $z_{\text{pre}}^c$ and $z_{\text{ft}}^c$ explores linearly evolving representation. (b) Linear interpolation between cross-Class Vectors with ViT-B/32 shows smooth transition between classes.

**Theorem 3.1** (CTL between pretrained and fine-tuned weights). *Suppose the function $f : \mathbb{R}^p \to \mathbb{R}$, and two fine-tuned weights $\theta_i$ and $\theta_j$ satisfy CTL [87]. Let $\theta_{\text{pre}}$ be the pre-trained weights. Define*

$$\delta_{\text{pre},i} = \left| f(\alpha\theta_{\text{pre}} + (1-\alpha)\theta_i) - \big((1-\alpha)f(\theta_{\text{pre}}) + \alpha f(\theta_i)\big) \right|,$$

$$\delta_{i,j} = \left| f(\alpha\theta_i + (1-\alpha)\theta_j) - \big((1-\alpha)f(\theta_i) + \alpha f(\theta_j)\big) \right|.$$

*If $\|\theta_i - \theta_{\text{pre}}\| < \|\theta_i - \theta_j\|$, then $\delta_{\text{pre},i} < \delta_{i,j}$: the segment from $\theta_{\text{pre}}$ to $\theta_i$ shows strictly smaller CTL deviation, hence is more linear, than the segment between two fine-tuned solutions $\theta_i \to \theta_j$.*

The proof is provided in Appendix §C.1. Using Theorem 3.1, we can apply CTL to pair of $\theta_{\text{pre}}$ and $\theta_{\text{ft}}$. Applying CTL to the set of inputs $x_c \in \mathcal{D}_c$ for a single class $c$ and averaging over $x_c$ yields:

$$f\big(x_c; \theta_{\text{pre}} + \alpha\tau\big) \approx f\big(x_c; \theta_{\text{pre}}\big) + \alpha\kappa_c, \qquad x_c \in \mathcal{D}_c.$$

Thus scaling the fixed vector $\kappa_c$ in latent space reproduces the effect of moving $\theta_{\text{pre}}$ along $\tau$ for class-$c$ samples, the class-specific adaptation. Fig. 2a (Top) visualizes this effect on ViT-B/32 across six downstream tasks: as $\alpha$ increases from 0 to 1, class-$c$ accuracy (normalized by the fully fine-tuned score) rises smoothly and concavely, confirming the linear path predicted by the theory. We observe similar behavior in both an MLP and ResNet-18 [22] with two other tasks (CIFAR10, CIFAR100), indicating that Class Vectors arise independently of network architecture or finetuning specifics (Fig. 2a (Middle), Fig. 2a (Bottom)). Experimental evidence for the inequality $\|\theta_i - \theta_{\text{pre}}\| < \|\theta_j - \theta_i\|$ and training details for all models are in Appendix Fig. 8 and §D.3.

*Take-away.* The adaptation required for a single class can be approximated by scaling a single latent vector $\kappa_c$; class-wise representation learning often reduces to simple vector arithmetic.

### 3.2 Class Vector-based Editing

To edit classifiers, we first construct an editing vector $z_{\text{edit}} \in \mathbb{R}^m$ in the latent space by linearly using the Class Vectors (§4). Prior work has shown that task vectors (in the weight space) and in-context vectors (in the latent space) can steer model behavior; our approach supports both injection modes.

**Latent-space injection.** Given $r = f(x, \theta_{\text{ft}}^e)$, we shift the representation by $z_{\text{edit}}$ and obtain $\hat{y} = g(r + z_{\text{edit}}, \theta^h)$. To avoid collateral edits in other classes (*i.e.*, to ensure localization of the edit), we gate the shift with $\beta = \mathbf{1}[\text{sim}(r) > \gamma]$, where $\text{sim}(r)$ is given by the cosine similarity to $z_{\text{ft}}^c$ and $\gamma$ denotes thresholds for gating (Algorithm.1). Please note that in most cases, $z_{\text{ft}}^c$ is known when constructing $z_{\text{edit}}$ (§4), and latent space injection does not require additional training.

**Weight-space mapping.** Latent-space manipulation of the classifier cannot fundamentally alter the deterministic rules encoded in the model's weights, leaving decision boundaries unchanged [48, 30]. It also imposes additional gating computations on the editor and requires maintaining $z_{\text{ft}}^c$. Following previous editing approaches [80, 60, 46] that embed edits directly into the model weights, we introduce a method for permanently embedding editing vectors into the model parameters. To mapping editing vectors in the weight space, we learn $\phi_{\text{edit}} : \mathbb{R}^m \to \mathbb{R}^{d_e}$ such that

$$\theta_{\text{edit}}^e = \theta_{\text{ft}}^e + \phi_{\text{edit}}(z_{\text{edit}}),$$

**Algorithm 1** Latent-space injection

1: **Inputs:** encoder $f(\cdot; \theta^e_{\text{ft}})$; head $g(\cdot; \theta^h)$; $x$; $z_{\text{edit}}$; class centroid $z^c_{\text{ft}}$; threshold $\gamma$
2: **Output:** logits $\hat{y}$
3: $r \leftarrow f(x; \theta^e_{\text{ft}})$
4: $\text{sim}(r) \leftarrow \frac{r^\top z^c_{\text{ft}}}{\|r\| \|z^c_{\text{ft}}\|}$
5: $\beta = \mathbf{1}[\text{sim}(r) > \gamma]$
6: $r_{\text{edit}} \leftarrow r + \beta\, z_{\text{edit}}$
7: $\hat{y} \leftarrow g(r_{\text{edit}}; \theta^h)$
8: **return** $\hat{y}$

**Algorithm 2** Weight-space mapping

1: **Input:** encoder $f(\cdot; \theta^e)$, data $\mathcal{X}_{\text{task}}$, $z_{\text{edit}}$, collecting number $N$, class $c$, epochs $T$
2: Freeze all but editable block $\mathcal{L}$
3: **for** $t = 1, \dots, T$ **and each** $(x, y) \in \mathcal{X}_{\text{task}}$ **do**
4: ⠀⠀Collect $N$ class-$c$ references $r_c = \{f(x) \mid y = c\}$
5: ⠀⠀Set $r_{\text{target}} = \text{mean}(r_c) + z_{\text{edit}}$ **if** $r_{\text{target}} = $ **None**
6: ⠀⠀$r = f(x)$ ; $\ell = \| r[y=c] - r_{\text{target}} \|^2$
7: ⠀⠀Update $\mathcal{L}$ with $\nabla_{\mathcal{L}} \ell$
8: **end for**
9: **return** $\theta^e_{\text{edit}}$

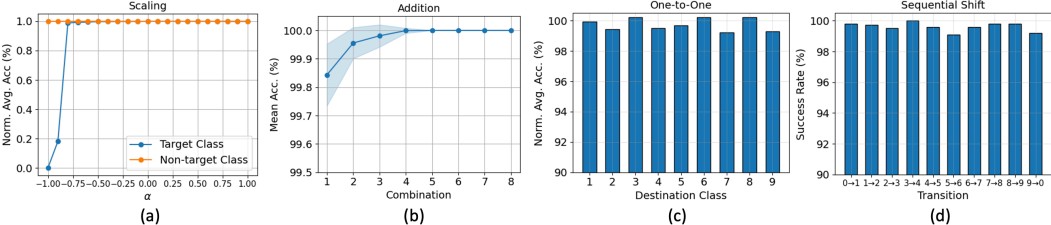

Figure 3: Independence of Class Vectors in MNIST. (a) Scaling the target class representation using $z_{\text{edit}} = \alpha \cdot \kappa_{c_1}$ (b) Adding non-target Class Vectors to the target class based on the combination count. (c) Modifying the target class to each destination class ($z_{\text{edit}} = \kappa_{\text{des.}} - \kappa_{\text{tar.}}$), with the averaged task accuracy. (d) Shifting all representations from $c_i \to c_{i+1}$ simultaneously with transition success rate.

minimising $\| f(x, \phi_{\text{edit}}(z_{\text{edit}})) - z_{\text{edit}} \|^2$. To make this more practical, assuming a linear mapping $f$ (a common assumption [81, 17]), the optimization reduces to $\theta^e_{\text{edit}}$:

$$\theta^e_{\text{edit}} = \underset{\theta^e_{\text{edit}}}{\arg\min} \| f(x, \theta^e_{\text{edit}}) - (f(x, \theta^e_{\text{ft}}) + z_{\text{edit}}) \|^2.$$

Similar to latent space steering, to ensure that the edit only affects the intended class, we first collect reference samples from that class and compute their latent representations. We then add the Class Vector $\kappa_c$ to each of these reference embeddings to form fixed target representations, and train the few encoder layers to map the original embeddings onto these shifted targets (Algorithm. 2). Theorem 3.2 shows that the mapping of Class Vectors from $\mathbb{R}^m$ into the model's weight space $\mathbb{R}^{d_e}$ admits infinitely many solutions, implying that it remains effective across diverse mapping configurations.

**Theorem 3.2** (Existence of a Mapping). *Let $\phi_{\text{edit}} : \mathbb{R}^m \to \mathbb{R}^{d_e}$ be any mapping that sends latent Class Vectors to weight perturbations applied in the encoder's final layer or a small subset of layers. Under the assumption that these edits are sufficiently small and confined to that small subset of layers of an overparameterized encoder (e.g., a ViT) with $d_e \gg m$, there exist infinitely many distinct $\phi_{\text{edit}}$.*

See Appendix §C.2 for a detailed proof. Note that Theorem 3.2 guarantees the existence of a valid mapping for overparameterized encoders, provided that the edit is restricted to a local subset of layers. This theoretical result suggests that such mappings may be inherently robust to the specific training procedure used for the encoder. The training setup for mapping are provided in Appendix §D.3.2.

### 3.3 Properties for Effective Classifier Editing

We now explore the properties of Class Vectors. Throughout our experiments, we employ the classes listed in Tab. 7 and validate our findings via the weight-space mapping approach.

**Linearity.** The linearity between two classes in a fine-tuned model is crucial for editing, as barriers or divergence along the path may cause $z_{\text{edit}}$ to fail, leading to unpredictable behavior. We interpolate between two fine-tuned classes $c_1$ and $c_2$ by $z_{\text{edit}} = -\alpha\, \kappa_{c_1} + \alpha\, \kappa_{c_2}$. This effectively shifts the model's adaptation from $c_1$ to $c_2$. As shown in Fig. 2b, predictions and logits change smoothly:

Table 1: Comparison of class unlearning with baselines, including the *mean* and *std* accuracies.

| Method | MNIST | | EuroSAT | | GTSRB | | RESISC45 | | DTD | |
|---|---|---|---|---|---|---|---|---|---|---|
| | $ACC_f$ ($\downarrow$) | $ACC_r$ ($\uparrow$) | $ACC_f$ ($\downarrow$) | $ACC_r$ ($\uparrow$) | $ACC_f$ ($\downarrow$) | $ACC_r$ ($\uparrow$) | $ACC_f$ ($\downarrow$) | $ACC_r$ ($\uparrow$) | $ACC_f$ ($\downarrow$) | $ACC_r$ ($\uparrow$) |
| Pretrained | 53.4±36.8 | 51.7±4.2 | 66.5±21.5 | 53.4±2.5 | 81.8±17.7 | 41.7±1.2 | 76.8±18.5 | 66.2±0.4 | 36.0±35.8 | 44.9±0.8 |
| Fine-tuned | 99.9±0.1 | 99.8±0.0 | 99.9±0.2 | 99.8±0.0 | 99.6±0.0 | 99.2±0.0 | 99.3±0.8 | 96.8±0.0 | 73.5±16.9 | 82.3±0.4 |
| Retrained | 0.1±0.1 | 76.4±0.0 | 0.0±0.0 | 85.7±0.0 | 41.8±25.3 | 57.5±0.0 | 33.9±20.6 | 75.0±0.0 | 14.5±26.5 | 55.5±0.0 |
| NegGrad | 0.0±0.0 | 43.4±10.3 | 0.0±0.0 | 11.6±1.0 | 0.0±0.0 | 15.6±25.6 | 0.0±0.0 | 2.5±0.6 | 0.0±0.0 | 13.6±16.7 |
| Random Vector | 99.9±0.1 | 99.8±0.0 | 99.9±0.1 | 80.9±26.1 | 99.6±0.5 | 98.2±0.7 | 100±0.0 | 79.4±19.1 | 72.0±31.8 | 51.1±11.1 |
| Class Vector | 0.0±0.0 | 99.7±0.0 | 0.0±0.0 | 99.5±0.2 | 0.0±0.0 | 98.6±0.0 | 28.2±26.1 | 94.6±7.2 | 13.5±16.5 | 78.1±0.8 |
| Class Vector$^\dagger$ | 0.0±0.0 | 96.2±0.1 | 0.0±0.0 | 99.7±0.0 | 0.0±0.0 | 93.4±0.0 | 10.0±10.9 | 90.7±3.2 | 15.2±18.7 | 72.9±0.1 |

samples switch cleanly from $c_1$ to $c_2$ at the midpoint, with no detours to other classes. This results show that Class Vectors permit precise linear edits of the classifier.

**Independence** To modify class $c_1$ towards $c_2$ without effect other classes, we require $f(x', \theta_{\text{ft}}^e) = f(x', \theta_{\text{ft}}^e + \phi(z_{\text{edit}}))$ for all $x' \notin c_1$, while $f(x_{c_1}, \theta_{\text{ft}}^e + \phi(z_{\text{edit}})) = f(x_{c_2}, \theta_{\text{ft}}^e)$. Neural Collapse (NC) phenomenon [55] states that, near the end of training, (i) all penultimate-layer features belonging to the same class tightly collapse to a class mean, and (ii) these class means themselves form a simplex Equiangular Tight Frame (ETF) centred at the global mean. Building on NC structure, we show that a class-specific update vector $\kappa_c$ exerts negligible influence on the embeddings of every other class.

**Theorem 3.3** (Independence of Class Vectors). *Suppose (i) the pretrained class embeddings collapse to a common mean $\bar{z}^{\text{pre}}$, that is $z_c^{\text{pre}} \approx \bar{z}^{\text{pre}}$; (ii) after fine-tuning the embeddings follow a centre-shifted ETF form $z_c^{\text{ft}} = \mu + u_c$ with $\sum_c u_c = 0$; and (iii) the global drift $\|\mu - \bar{z}^{\text{pre}}\|$ is negligible compared to the class-specific update $\|u_c\|$. Then, for any two distinct classes $c \neq c'$,*

$$\cos(\kappa_c, z_{c'}^{\text{ft}}) \approx 0,$$

*i.e. the Class Vector $\kappa_c$ is approximately orthogonal to the fine-tuned embedding of every other class.*

A detailed proof and empirical evidence that strongly support these conditions for ViT are presented in Appendix §C.3. In Fig. 3, we empirically evaluate the independence of Class Vectors. It shows that Class Vectors preserve the accuracy of non-target classes and ensure independent edits across classes, even when multiple classes are edited simultaneously.

## 4 Editing Classifiers

We now introduce editing applications with Class Vectors. For all applications, we first design $z_{\text{edit}}$, then steer the models in latent spaces or map it to the weight space to alter their predictive behavior.

### 4.1 Experimental Setups

To evaluate Class Vectors for classifier editing, we extract pretrained and fine-tuned class centroids from three widely adopted CLIP encoders, ViT-B/16, ViT-B/32, and ViT-L/14 [58], and form Class Vectors as their differences. In §4.3 and §4.4, Class Vectors are derived from initialized and pretrained encoders, as they predict ImageNet classes, the pretraining dataset for these classifiers. We denote latent-space steering by *Class Vector* and weight-space mapping by *Class Vector*$^\dagger$, using a default similarity threshold of $\gamma = 0.5$. As baselines, we include *Retrained* [80, 60], which retrains only the target class using cross-entropy loss (or excludes it entirely in the unlearning setting), and *Random Vector*, which is initialized to match the magnitude of $z_{\text{edit}}$ and mapped to the weight space to test for non-semantic effects. Among all considered methods, only the *Class Vector* method enables latent steering without requiring any additional training.

We note that *Task Vector* [29] is not included as a baseline in our main experiments, since it operates at the task-wide level and is unsuitable for evaluating class-wise editing. For completeness, we additionally provide comparative results between task vectors and Class Vectors in the class unlearning setting in Tab. 17. Additional task-specific baselines are described in their respective sections, and full experimental details in Appendix §D.

## 4.2 Class Unlearning

Since ViT classifiers are exposed to a wide range of data during training, they naturally encode information across all classes. Practically, class unlearning is intended to modify classifier decision boundaries to prevent classification into specific categories, while minimizing unintended changes to non-target classes, often motivated by privacy or security concerns [52]. Existing class unlearning methods typically involve retraining the model or performing multi-stage post hoc edits, which are computationally expensive and risk inadvertently erasing features shared across classes [7].

To evaluate Class Vectors on class unlearning, we set $z_{\text{edit}} = \lambda \cdot \kappa_c$ with $\lambda = -1.5$ to effectively modify the model to unlearn adapted predictive rules for class $c$. We utilize the ViT-B/16 model for our experiments (See Appendix §E for results on other ViTs). For an additional baseline method, we retrain models with gradient ascent to the target class (*i.e.*, , NegGrad) following previous studies [37, 40]. For mapping, we utilize the single reference samples in the subset of each task's test set and evaluate on the remaining data, training only the final layer's layer normalization of the encoder. The first five labels in each task are used as the target (forget) class in each experiment. As shown in Tab. 1, Class Vectors demonstrate the most effective editing strategy for unlearning the target class ($ACC_f$) while preserving the performance of the non-target classes ($ACC_r$). In contrast, random vectors have minimal effects, indicating that Class Vectors point to meaningful directions in the latent space. Additionally, retraining struggles with limited data and trainable parameters.

## 4.3 Adapting to New Environment

Imbalanced training scenes often lead classifiers to reduced performance in specific contexts [65]. Thus, adapting open-hub classifiers to suit individual users' environments in an efficient manner is necessary in practice. Following a previous study [60], we examine a scenario where the model struggles to classify objects in snowy environments (Fig. 4). This occurs when the representation of snowy objects fails to accurately capture the object's features due to the ambiguous influence of snow.

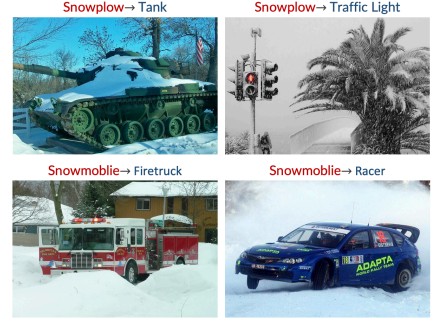

Class Vectors can effectively address this through high-level concept-based arithmetic operations. Specifically, our goal is to eliminate the snow features from the image representation. Therefore, the objective of $z_{\text{edit}}$ is:

$$c_{\text{object}} = g(z^{\text{object}}, \theta^h) = g(z^c + z_{\text{edit}}), \theta^h)), \quad (1)$$

Figure 4: Adapting the classifier to a snowy environment. Red text marks the misclassifications made by the original model, while blue text shows the correct predictions after classifier editing.

where $z_{\text{edit}}$ is designed to eliminate the representation activated by the model when snow is input, ensuring that the model focuses solely on the object.

We consider practical scenarios when only limited external samples are accessible. Namely, the editor can obtain a few images of the target object $c_1$ from external sources. We set $z_{\text{edit}}$ as $\lambda(\kappa_{\text{snow}+c_1} - \kappa_{c_1})$. Here, $\kappa_{\text{snow}+c_1}$ denotes the representation adaptation of the model with snowy images. At a high level, this retains only the model adaptation related to snow. Editor can steer the model or map $z_{\text{edit}}$ to satisfy Eq. 1, with $\lambda < 0$ used to suppress snow features. We evaluate on Snowy ImageNet [60] (7 classes, 20 images each), using 5 reference samples per class for mapping and testing on the remainder for mapping experiments. As an additional baseline, DirMatch [60] trains models to align images to target-class representations using external samples individually. We train only the final MLP and layer-norm in the transformer block, reporting mean ± standard deviation accuracy across classes. With $\lambda = -1.0$ and 4 external samples (Fig. 10), Class Vectors deliver a 10–20% improvement over the pre-edit classifier (Tab. 2a), underscoring their high-level interactions and effectiveness.

## 4.4 Defending Against Typography Attacks

Vision-language pretrained models like CLIP have exhibited vulnerabilities to typography attacks, where the text in an image leads to misclassification of the model [16]. Thus, mitigating typographic attack risk is crucial in safety-critical domains, such as medical imaging, before deploying classifiers. Similar to §4.3, given an object with text written on it, the representation is expected to become

Table 2: Results on classifier editing with Class Vectors in two scenarios: (a) adapting to a new distribution (snowy environments) and (b) defending against typographic attacks.

| Method | Average (↑) | | |
|---|---|---|---|
| | ViT-B/16 | ViT-B/32 | ViT-L/14 |
| Pretrained | 55.2±24.6 | 53.4±29.9 | 60.2±22.5 |
| Retrained | 55.8±48.4 | 55.8±48.5 | 75.3±15.5 |
| Random Vector | 26.2±23.0 | 16.2±22.7 | 49.7±26.1 |
| DirMatch | 72.0±22.8 | 73.9±23.5 | 74.6±16.4 |
| Class Vector | 69.7±2.6 | 72.2±3.8 | 71.3±3.2 |
| Class Vector$^{\dagger}$ | 72.7±21.4 | 76.2±21.2 | 78.3±16.9 |

(a)

| Method | Average (↑) | | |
|---|---|---|---|
| | ViT-B/16 | ViT-B/32 | ViT-L/14 |
| Pretrained (Clean) | 75.0±38.2 | 100±0.0 | 100±0.0 |
| Pretrained (Attack) | 48.9±38.2 | 76.7±33.1 | 38.9±19.4 |
| Retrained | 80.0±34.2 | 66.7±47.1 | 98.8±2.5 |
| Random Vector | 41.1±30.7 | 74.4±38.6 | 33.3±21.4 |
| DirMatch | 97.7±3.1 | 91.1±8.3 | 87.8±13.0 |
| Class Vector | 88.9±22.0 | 98.9±2.5 | 93.3±7.7 |
| Class Vector$^{\dagger}$ | 98.9±2.5 | 99.0±2.5 | 93.3±6.7 |

(b)

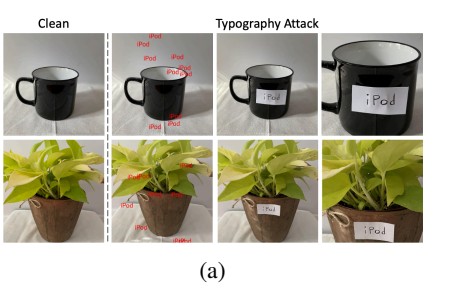

(a)

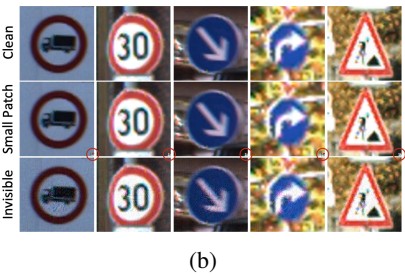

(b)

Figure 5: Visual examples for the scenarios in §4.4 and §4.5: (a) examples of typographic attacks that cause the model to misclassify inputs as iPod. (b) optimized backdoor triggers on traffic-sign images.

$z^{\text{object}}$ after classifier editing, allowing the deterministic rules of the model to focus solely on the object for accurate recognition. Thus, our goal aligns with that described in Eq. 1. Practically, the editor can easily generate small set of augmented samples by directly adding text to objects or performing data augmentation. Using them, the Class Vectors $\kappa_{\text{text+object}}$ for text-affected images and $\kappa_{\text{object}}$ for clean images can be derived by feeding each image sets into the model. Finally, we define $z_{\text{edit}} = \lambda(\kappa_{\text{text+object}} - \kappa_{\text{object}})$, where $z_{\text{edit}}$ removes adaptations from text-object images at a high-level, isolating the model's clean object representations.

Following a previous study [60], we utilize web-sourced image sets with 6 classes from ImageNet that include both clean and text with objects. Each image is augmented into 15 images per class (Fig. 5a). For mapping, we use a total of 4 reference images for training and evaluate on the remaining images. We use the same baselines as §4.3, setting $\lambda = -1.5$. For DirMatch, text-augmented images are trained to directly align with clean image representations. Results in Tab. 2b demonstrate that the Class Vector effectively defends against typographic attacks, achieving average accuracies on par with or exceeding clean-image performance across models.

## 4.5 Adversarial Trigger Optimizations

Class Vectors can also be utilized for backdoor attacks, which alter a model's logic using adversarial triggers such as small patches [20] or imperceptible noise [85], typically optimized by training the classifier to misclassify triggered images [82]. However, primary limitations of these approaches are the requirement for large amounts of triggered samples and full access to the training process to adjust the model's weights. We consider a scenario where an editor (*i.e.*, attacker) aims to mislead a classifier into misclassifying specific classes, without altering the model's weights. The attacker knows the classifier's architecture but lacks access to the user's model or training process. With knowledge of the architecture, the attacker can acquire a model with the same design from an open hubs. By embedding the intended representation into a trigger patch or an invisible trigger using the same model architecture, the attacker can cause the classifier to misclassify any object or scene where the pattern appears, whether on the object itself or attached to the camera lens.

To effectively embed malicious representations into the trigger, the attacker aims to optimize the initialized trigger $x_{\text{trigger}}$'s representation into $z_{\text{edit}} = \lambda \cdot (\kappa_{c_2} - \kappa_{c_1})$ such that when $x_{\text{trigger}}$ is attached to an image, the classifier misclassifies $c_1$ as $c_2$. Unlike previous sections, $z_{\text{edit}}$ is mapped to the pixel

space by training the pixels in $x_{\text{trigger}}$ (Algorithm 4):

$$x_{\text{trigger}} = \underset{x_{\text{trigger}}}{\arg\min} \, \| f(x_{\text{trigger}}, \theta_{\text{ft}}^e) - z_{\text{edit}} \|^2. \tag{2}$$

Note that, because the backdoor triggers are optimized, latent-space steering cannot be applied in this scenario. Class Vectors are evaluated across real-world tasks, including GTSRB, RESISC45, and SVHN. For mapping, 30, 10, and 200 samples are selected from each test set, with performance assessed on the remaining data. We additionally include BadNet [20] for baselines, where weights are manipulated with unlearnable triggered images to classify into the destination class.

We measure Attack Success Rate (ASR) on triggered images and Clean Accuracy (CA) on clean images, averaged across tasks. For DirMatch, the triggers are optimized directly toward the destination representation of target class. The scaling coefficient $\lambda$ for $z_{\text{edit}}$ is set to 1.5 for small patches, using 0.8% of total pixels and 1.0 for invisible noise by default. All experiments use ViT-B/32 (see Appendix §E for results on other ViTs). Tab. 3 shows that Class Vectors achieve high ASR, achieving high effectiveness without modifying model weights. As shown in Fig. 5b, triggers are optimized to be either very small or stealthy, making them imperceptible.

Table 3: Results on backdoor attacks with optimized triggers.

| Method | Small Patch | | Invisible | |
|---|---|---|---|---|
| | ASR (↑) | CA (↑) | ASR (↑) | CA (↑) |
| Pretrained | 19.6±26.2 | 43.8±9.7 | 22.2±27.9 | 43.8±9.7 |
| Finetuned | 0.0±0.1 | 95.2±7.1 | 0.0±0.1 | 95.2±7.1 |
| BadNet | 100±0.0 | 10.0±5.0 | 100±0.0 | 9.3±7.4 |
| Random Vector | 0.1±0.1 | 95.2±7.1 | 39.4±55.8 | 95.2±7.1 |
| DirMatch | 96.5±4.7 | 95.2±7.1 | 96.8±4.5 | 95.2±7.1 |
| Class Vector† | 99.8±2.8 | 95.2±7.1 | 99.0±1.4 | 95.2±7.1 |

Table 4: Results of class unlearning on MNIST using ResNet18, ResNet50, and ConvNeXT-Tiny.

| Method | ResNet18 | | ResNet50 | | ConvNeXT-Tiny | |
|---|---|---|---|---|---|---|
| | $ACC_f$ (↓) | $ACC_r$ (↑) | $ACC_f$ (↓) | $ACC_r$ (↑) | $ACC_f$ (↓) | $ACC_r$ (↑) |
| Retrained | 99.8 | 99.5 | 89.9 | 99.6 | 78.2 | 99.4 |
| NegGrad | 14.2 | 97.2 | 1.0 | 95.0 | 0.0 | 11.2 |
| Random Vector | 99.7 | 99.4 | 99.5 | 99.4 | 99.5 | 99.3 |
| Class Vector | 2.2 | 97.0 | 11.5 | 84.2 | **0.0** | 95.3 |
| Class Vector† | **0.0** | **99.4** | **0.0** | **99.1** | **0.0** | **99.1** |

Table 5: Class unlearning with BERT-Base.

| Method | AG-NEWS | | DBPedia-14 | | 20-Newsgroups | |
|---|---|---|---|---|---|---|
| | $ACC_f$ (↓) | $ACC_r$ (↑) | $ACC_f$ (↓) | $ACC_r$ (↑) | $ACC_f$ (↓) | $ACC_r$ (↑) |
| Retrained | 71.9 | 89.3 | 98.4 | 99.0 | 59.8 | 66.5 |
| NegGrad | 0.0 | 48.9 | 0.0 | 93.8 | 0.0 | 47.0 |
| Random Vector | 93.8 | 94.3 | 98.6 | 99.1 | 62.9 | 67.9 |
| Class Vector | 0.0 | 93.2 | 0.0 | 96.9 | 0.0 | 57.9 |
| Class Vector† | **3.2** | **94.4** | **0.0** | **99.1** | **0.0** | **63.8** |

### 4.6 In-Depth Analysis

**Model Generality across Architectures.** To examine the architectural generality of Class Vectors, we extend our analysis beyond ViT encoders to convolutional and language models, including ResNet18, ResNet50, ConvNeXT-Tiny [45], and BERT-Base [12]. For each model, Class Vectors are derived from the pretrained and fine-tuned representations and applied to the class unlearning setting, targeting the first class in the dataset. As shown in Tab. 4 and Tab. 5, Class Vector maintains strong forgetting performance while preserving non-target accuracy across all architectures. Notably, both the latent-space variant (*Class Vector*) and its weight-space mapping (*Class Vector*†) exhibit consistent gains compared to gradient-based or random baselines, confirming that Class Vectors capture transferable, semantically meaningful directions independent of network type.

**Impact of threshold in latent space steering.** In the latent-space steering method, Class Vector injection is applied only to inputs whose cosine similarity to the target feature exceeds the threshold $\gamma$. To examine the impact of $\gamma$, we perform a systematic $\gamma$ sweep accompanied by class-unlearning

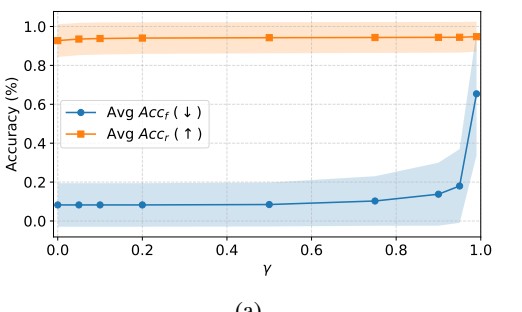 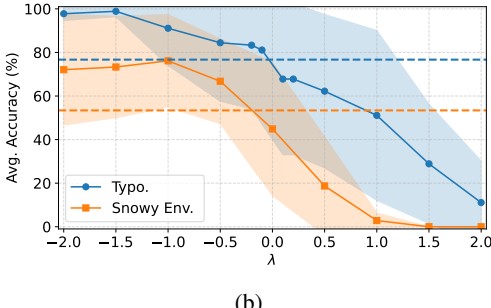

| (a) | (b) |

Figure 6: In-depth analysis: (a) Effect of the cosine-similarity threshold $\gamma$ on class-unlearning clean accuracy; (b) Effect of the scaling coefficient $\lambda$ on controllable editing. Horizontal dashed lines denote pretrained model performance.

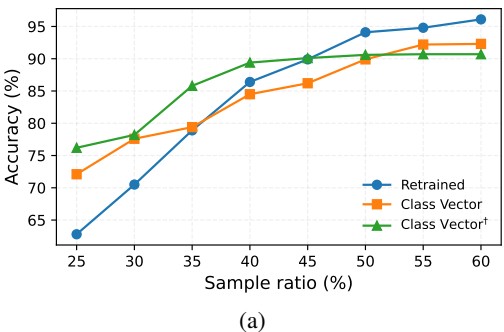 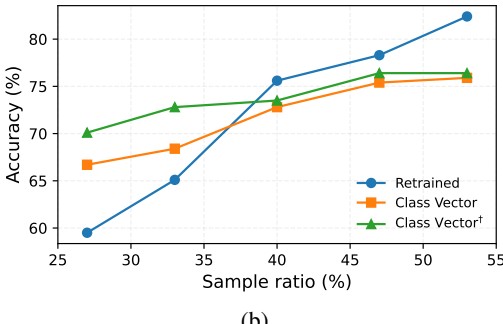

| (a) | (b) |

Figure 7: Effect of sample ratio on editing performance. Each curve shows how model accuracy varies as the proportion of available samples increases for (a) Snowy ImageNet and (b) Typo Attack.

evaluations to assess both editing efficacy and collateral impact. In Fig. 6a, editing remains effective for $\gamma \in [0.0, 0.5]$, and, even at $\gamma = 0$, we observe that edits retain independence, owing to the independence of Class Vectors (Theorem 3.3).

**Impact of scaling coefficient.** The scaling coefficient $\lambda$ determines both the strength and direction of an edit. To make editing intuitive for non-experts, $\lambda$ should yield predictable, controllable outcomes. For instance, in defense against typographic attacks, a more negative $\lambda$ aggressively removes the learned "iPod" adaptation, while in other cases a milder edit suffices. We sweep $\lambda$ in the snowy-environment and typography-attack (Fig. 6b) scenarios, observing clear trends: negative values erase snow features and restore correct predictions, whereas positive values amplify them and degrade performance. This demonstrates that $\lambda$ can be tuned reliably based on high-level editing goals.

**Impact of sample size.** Retraining-based methods improve with more samples, but such abundance is rare in real deployments. To test scalability, we varied the sample ratio—the portion of available target data—in Snowy ImageNet and Typo Attack (Fig. 7a, 7b). With less than 30% of data, both Class Vector and Class Vector$^{\dagger}$ outperform retraining, which only catches up after 35%. This highlights Class Vector's strong data efficiency, leveraging latent semantic directions instead of full parameter optimization, and maintaining competitiveness even under low-data regimes.

## 5 Conclusion and Future Work

We have introduced Class Vectors, which capture class-specific representation adaptations during training. Open-hub models can leverage Class Vectors to modify predictive rules for task-specific personalization. Our analysis of their linearity and independence, supported by extensive experiments, highlights their potential for efficient and interpretable classifier editing across diverse applications. While our work primarily focuses on image classification, we anticipate future extensions of Class Vectors to natural language processing (NLP) and generative models, including Large Language Models (LLMs) and image generative models.

## Acknowledgements

This work was supported in part by National Research Foundation of Korea (NRF) grant (RS-2025-00560762), and Institute of Information & communications Technology Planning & Evaluation (IITP) grant (RS-2025-02263754, RS-2025-25442338, IITP-2025-RS-2024-00397085, RS-2021-II211343). This research was also conducted as part of the Sovereign AI Foundation Model Project (Data Track, 2025-AI Data-wi43), organized by the Ministry of Science and ICT (MSIT) and supported by the National Information Society Agency (NIA). J. Do is with ASRI, Seoul National University.

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

# Supplementary Material

## Contents

# A Broader Discussion

## A.1 Extended Related Work

**Model interventions**   Model intervention aims to adapt trained models to new knowledge for specific user needs. Retraining an entire model is time-consuming and data-intensive, driving the development of more efficient methods that use limited data. These include model alignment [1, 54, 70, 32], debugging [59], and editing [29, 60, 50, 4]. Model intervention in Computer Vision (CV) remains limited. Previous work propose editing generative adversarial networks (GANs) by identifying and modifying specific locations [4], while this approach is extended to classifiers for debugging errors by mapping new rules to existing ones [60]. However, these methods require editing locations or additional data. More recent meta-learning-based approach [80] address this but remain computationally demanding.

**Latent representations**   Latent space interpolation in generative models aims to blend the styles of different images by navigating their latent vectors. This approach has been widely explored in GANs [69, 31] and diffusion models [71, 2], while its application to classifiers remain largely unexplored. Meanwhile, recent efforts to explain deep neural networks have focused on linking models' internal processes to high-level concepts, such as analyzing human-understandable features or testing their influence on predictions (*i.e.*, decomposability) [8, 14, 34, 3, 19]. Building on these ideas, we investigate representation adaptation in the latent space, revealing properties and enabling effective, high-level editing.

## A.2 Limitations

While Class Vectors enable efficient and interpretable classifier editing across a variety of tasks, our method exhibits several limitations. First, the approach assumes that class representations are well-structured and approximately linearly separable in the latent space. This assumption is empirically supported by CTL and Neural Collapse, but may not hold in scenarios with high intra-class variance, noisy labels, or long-tailed distributions. Second, the method currently focuses on single-label classification tasks where each example is associated with a single semantic class. Extending Class Vector-based editing to multi-label or hierarchical classification, where class boundaries are less distinct and often overlapping, remains an open challenge. In addition, the latent-space steering method relies on access to the centroid representation of each class, which may not be readily available in privacy-constrained or black-box settings. Similarly, the weight-space mapping method requires updating a small subset of layers with a few reference samples, which still assumes partial access to model internals. This limits the applicability of our method in fully closed-source environments. Despite these limitations, our work lays the foundation for structured classifier editing and invites further research into expanding its scope to more complex, unconstrained settings.

## A.3 Ethic Statement

This work does not involve research with human participants, sensitive data, or personally identifiable information. All experiments were conducted using publicly available pretrained models and open-source datasets. We include limited web-sourced or user-generated imagery (e.g., for typographic attack evaluation), and such images are used solely for non-commercial, academic research purposes under fair use or Creative Commons–compliant terms. While our method enables editing of classifiers for beneficial purposes such as unlearning and robustness, we recognize that similar techniques may be repurposed for malicious intent, such as backdoor trigger optimization. To mitigate such risks, we release code and dataset under a non-commercial license (CC-BY-NC-SA 4.0) and emphasize that practical deployment of editing techniques should be preceded by careful threat modeling and access control.

## A.4 Licenses

We plan to release our code under the Apache 2.0 license.

# B  Algorithms

We present algorithms that leverage Class Vectors by mapping them into non-latent spaces. For latent space steering, refer to Algorithm 1. Specifically, we introduce two approaches: (1) standard mapping via an encoder, and (2) pixel-space mapping for adversarial trigger optimization.

---

**Algorithm 3** Pseudocode for optimizing classifier encoder with Class Vector

---

**Require:** Classifier encoder ($f(\cdot, \theta^e)$ and trainable layers), Dataset $\mathcal{X}_{\text{task}}$, Editing vector $z_{\text{edit}}$, Number of epochs $T$, Target class $c$, Learning rate $\eta$
**Ensure:** Edited encoder weight $\theta^e_{\text{edit}}$ such that $f(x \in \mathcal{X}_{\text{task}}, \theta^e) = z^c + z_{\text{edit}}$
 1: **Freeze** all layers of $\theta^e$ **except** the final trainable layers
 2: $\mathbf{r}_{\text{target}} \leftarrow$ None
 3: **Initialize** list $\mathcal{R} \leftarrow [\,]$    {to store class-$c$ representations}
 4: **for** epoch $= 1$ to $T$ **do**
 5:    **for** each mini-batch $(X, Y)$ in $\mathcal{X}$ **do**
 6:       $\mathbf{r} \leftarrow f(X, \theta^e)$    {Encoder representation}
 7:       **if** $\mathbf{r}_{\text{target}} =$ None **then**
 8:          **Collect** $\mathbf{r}_c \leftarrow \mathbf{r}[Y = c]$, **Append** $\mathbf{r}_c$ to $\mathcal{R}$    {Representations for target class}
 9:          **if** enough class-$c$ reps in $\mathcal{R}$ **then**
10:             $\bar{\mathbf{r}} \leftarrow \text{mean}(\mathcal{R})$    {Average representation **when reference sample number is satisfied**}
11:             $\mathbf{r}_{\text{target}} \leftarrow \bar{\mathbf{r}} + z_{\text{edit}}$    {Add editing vector}
12:          **end if**
13:          **Continue to next mini-batch if** $\mathbf{r}_{\text{target}} =$ None
14:       **end if**
15:       **Filter** $\widetilde{\mathbf{r}} \leftarrow \mathbf{r}[Y = c]$    {Only align class-$c$ representations}
16:       **Compute alignment loss:** $\ell = \|\widetilde{\mathbf{r}} - \mathbf{r}_{\text{target}}\|^2$
17:       **Backpropagation with** $\ell$
18:    **end for**
19: **end for**
20: **return** $M$    {Edited encoder weight $\theta^e_{\text{edit}}$}

---

---

**Algorithm 4** Pseudocode for optimizing adversarial trigger with Class Vector

---

**Require:** Classifier encoder $f(\cdot, \theta^e)$, Trainable trigger $x_{\text{trigger}}$, Dataset $\mathcal{X}_{\text{task}}$, Editing vector $z_{\text{edit}}$, Number of epochs $T$, Target class $c$, Learning rate $\eta$
**Ensure:** Edited encoder weight $\theta^e_{\text{edit}}$ such that $f(x + x_{\text{trigger}}, \theta^e) = z^c + z_{\text{edit}}$
    $\mathbf{r}_{\text{target}} \leftarrow$ None
 2: **Initialize** list $\mathcal{R} \leftarrow [\,]$    {to store class-$c$ representations}
    **for** epoch $= 1$ to $T$ **do**
 4:    **for** each mini-batch $(X, Y)$ in $\mathcal{X}$ **do**
       $\mathbf{r} \leftarrow f(X + x_{\text{trigger}}, \theta^e)$    {Encoder representation}
 6:       **if** $\mathbf{r}_{\text{target}} =$ None **then**
          **Collect** $\mathbf{r}_c \leftarrow \mathbf{r}[Y = c]$, **Append** $\mathbf{r}_c$ to $\mathcal{R}$    {Representations for target class}
 8:          **if** enough class-$c$ reps in $\mathcal{R}$ **then**
             $\bar{\mathbf{r}} \leftarrow \text{mean}(\mathcal{R})$    {Average representation **when reference sample number is satisfied**}
10:             $\mathbf{r}_{\text{target}} \leftarrow \bar{\mathbf{r}} + z_{\text{edit}}$    {Add editing vector}
          **end if**
12:          **Continue to next mini-batch if** $\mathbf{r}_{\text{target}} =$ None
       **end if**
14:       **Filter** $\widetilde{\mathbf{r}} \leftarrow \mathbf{r}[Y = c]$    {Only align class-$c$ representations}
       **Compute alignment loss:** $\ell = \|\widetilde{\mathbf{r}} - \mathbf{r}_{\text{target}}\|^2$
16:       **Backpropagation with** $\ell$
    **end for**
18: **end for**
    **return** $x_{\text{trigger}}$    {Edited trigger $x_{\text{trigger}}$}

---

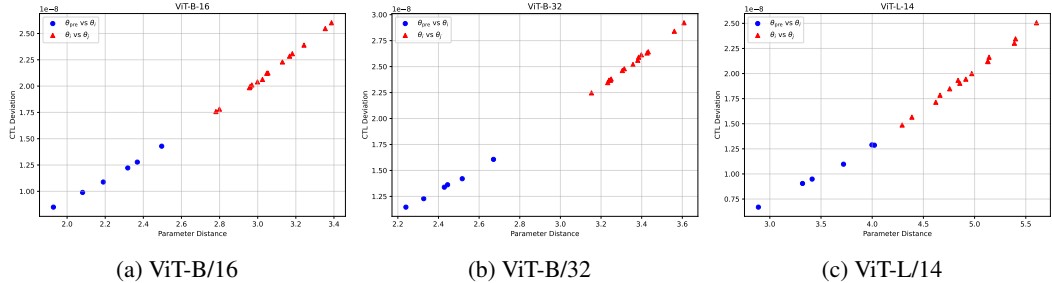

(a) ViT-B/16      (b) ViT-B/32      (c) ViT-L/14

Figure 8: Empirical validation of Theorem 3.1: CTL deviation increases with parameter distance. Blue markers denote interpolation between pretrained and fine-tuned weights ($\theta_{\mathrm{pre}} \leftrightarrow \theta_i$); red markers denote interpolation between different fine-tuned weights ($\theta_i \leftrightarrow \theta_j$).

## C  Supplementary Theoretical Analysis

In this section, we provide theoretical analyses, including proofs for the theorems, as well as empirical evidence supporting them.

### C.1  Theoretical Justification of Class Vectors

Here, we prove Theorem 3.1 and provide empirical results supporting the assumptions illustrated in the figure.

**Theorem 3.1** (CTL between pretrained and fine-tuned weights). *Suppose the function $f : \mathbb{R}^p \to \mathbb{R}$ is three-times differentiable on an open convex set $\Theta \subset \mathbb{R}^p$, and that its Hessian is spectrally bounded at every $\theta_0 \in \Theta$: $\lambda_{\min} \leq \left\| \nabla^2 f(\theta_0) \right\| \leq \lambda_{\max}$ [87]. Let $\theta_{\mathrm{pre}}$ be the pre-trained weights and $\theta_i, \theta_j$ two fine-tuned weights that satisfy CTL. Define*

$$\delta_{\mathrm{pre},i} = \left| f(\alpha\theta_{\mathrm{pre}} + (1-\alpha)\theta_i) - \big((1-\alpha)f(\theta_{\mathrm{pre}}) + \alpha f(\theta_i)\big) \right|,$$
$$\delta_{i,j} = \left| f(\alpha\theta_i + (1-\alpha)\theta_j) - \big((1-\alpha)f(\theta_i) + \alpha f(\theta_j)\big) \right|.$$

*If $\|\theta_i - \theta_{\mathrm{pre}}\| < \|\theta_i - \theta_j\|$, then $\delta_{\mathrm{pre},i} < \delta_{i,j}$: the segment from $\theta_{\mathrm{pre}}$ to $\theta_i$ shows strictly smaller CTL deviation, hence is* more linear*, than the segment between two fine-tuned solutions $\theta_i \to \theta_j$.*

*Proof.* From Theorem 5.1 in *Zhou et al.* [87], for any pair $\theta_a, \theta_b \in \Theta$, if $f : \mathbb{R}^p \to \mathbb{R}$ is three-times differentiable on open convex domain $\Theta$, and its Hessian satisfies the spectral bound

$$\lambda_{\min} \leq \|\nabla^2 f(\theta)\| \leq \lambda_{\max}, \quad \forall \theta \in \Theta,$$

then the CTL deviation is bounded by:

$$\delta_{\theta_a,\theta_b} = |f(\alpha\theta_a + (1-\alpha)\theta_b) - [\alpha f(\theta_a) + (1-\alpha)f(\theta_b)]| \leq \frac{\alpha(1-\alpha)}{2}\lambda_{\max}\|\theta_a - \theta_b\|^2 + E,$$

where the remainder term $E = \mathcal{O}(\|\alpha\theta_a + (1-\alpha)\theta_b - \theta_0\|^3)$ vanishes as the interpolation point approaches $\theta_0$.

Now fix $\theta_{\mathrm{pre}}, \theta_i, \theta_j \in \Theta$, and assume:

$$\|\theta_i - \theta_{\mathrm{pre}}\| < \|\theta_i - \theta_j\|.$$

Apply the bound to each deviation:

$$\delta_{\mathrm{pre},i} \leq \frac{\alpha(1-\alpha)}{2}\lambda_{\max}\|\theta_i - \theta_{\mathrm{pre}}\|^2 + E_1,$$

$$\delta_{i,j} \leq \frac{\alpha(1-\alpha)}{2}\lambda_{\max}\|\theta_i - \theta_j\|^2 + E_2,$$

for small remainder terms $E_1, E_2$ of order $\mathcal{O}(\|\cdot\|^3)$.

Because $\|\theta_i - \theta_{\mathrm{pre}}\| < \|\theta_i - \theta_j\|$, it follows that:
$$\delta_{\mathrm{pre},i} < \delta_{i,j} \quad \text{up to a cubic-order error.}$$

Therefore, the CTL deviation along the interpolation path from $\theta_{\mathrm{pre}}$ to $\theta_i$ is strictly smaller than that between $\theta_i$ and $\theta_j$ in the small-distance regime. $\qquad\square$

Fig. 8 illustrates the empirical validation of Theorem 3.1, which states that the CTL deviation between two parameter vectors is upper-bounded by a quadratic function of their Euclidean distance. CTL deviations are computed using a synthetic quadratic loss function as a surrogate for the true task loss. In particular, the deviation tends to be smaller when interpolating between a pretrained model and a fine-tuned model ($\theta_{\mathrm{pre}} \leftrightarrow \theta_i$) than between two independently fine-tuned models ($\theta_i \leftrightarrow \theta_j$).

## C.2 Existence of a Mapping Function

The mapping function $\phi_{\mathrm{edit}}$ effectively exists in practical editing scenarios, such as small weight modifications within an overparameterized encoder. We demonstrate that KL-divergence–based mapping also performs effectively (Tab. 19). Furthermore, experiments across various learning rate configurations (Tab. 18) show that the mapping is robust to different optimization setups.

**Theorem 3.2** (Existence of a Mapping). *Let $\phi_{\mathrm{edit}} : \mathbb{R}^m \to \mathbb{R}^{d_e}$ be any mapping that sends latent editing vectors to weight perturbations applied in the encoder's final layer or a small subset of layers. Under the assumption that these edits are sufficiently small and confined to that small subset of layers of an overparameterized encoder (e.g., a ViT) with $d_e \gg m$, there exist infinitely many distinct $\phi_{\mathrm{edit}}$.*

*Proof.* For any input $x$, if the editable–parameter perturbation $w$ is sufficiently small, then
$$f(x; \theta + w) = f(x; \theta) + J_f(\theta)w + o(\|w\|),$$
so a first–order Taylor approximation around $\theta$ is valid. Let $J := J_f(\theta) \in \mathbb{R}^{m \times d_e}$ denote the Jacobian of the encoder output with respect to the editable parameters, evaluated at $\theta$. Since edits are restricted to an overparameterised subset of layers with $d_e \gg m$, we assume $\mathrm{rank}(J) = m$ (full row rank).

We seek a perturbation $w \in \mathbb{R}^{d_e}$ that realises a target change $z \in \mathbb{R}^m$ in the encoder output, i.e.
$$Jw = z.$$
Because $J$ has full row rank, the matrix $JJ^\top \in \mathbb{R}^{m \times m}$ is invertible, and the Moore–Penrose pseudoinverse
$$J^\dagger := J^\top (JJ^\top)^{-1} \in \mathbb{R}^{d_e \times m}$$
satisfies $JJ^\dagger = I_m$. Hence one particular solution of $(A)$ is
$$w_0(z) := J^\dagger z.$$

Let $\mathcal{N} := \ker J = \{v \in \mathbb{R}^{d_e} \mid Jv = 0\}$. Its dimension is $d_e - m > 0$, so the full solution set of $(A)$ is the affine subspace
$$\mathcal{S}(z) = J^\dagger z + \ker J = \{\, J^\dagger z + v \mid v \in \ker J \,\}.$$
Since $\mathcal{N}$ is nontrivial, infinitely many distinct $w$ satisfy $Jw = z$.

Now take any linear map $N \in \mathbb{R}^{d_e \times m}$ whose columns lie in $\ker J$ ($JN = 0$), and define
$$R := J^\dagger + N.$$
Then $JR = JJ^\dagger = I_m$, so for every $z \in \mathbb{R}^m$,
$$J(Rz) = z.$$
The mapping
$$\phi_{\mathrm{edit}}(z) := Rz$$
thus provides a valid weight–space perturbation realising the desired edit. Varying $N$ (or equivalently adding any vector in $\ker J$ to $Rz$) yields infinitely many distinct mappings $\phi_{\mathrm{edit}} : \mathbb{R}^m \to \mathbb{R}^{d_e}$ that all satisfy $J\phi_{\mathrm{edit}}(z) = z$.

Finally, since $\ker J$ is a linear subspace, for any $\varepsilon > 0$ we may rescale the target $z$ (or equivalently $R$) to ensure $\|\phi_{\mathrm{edit}}(z)\| \le \varepsilon$, so the perturbation remains sufficiently small while achieving the desired first–order effect. Therefore, under the stated conditions, there exist infinitely many sufficiently small weight–space mappings $\phi_{\mathrm{edit}}$ that satisfy $J\phi_{\mathrm{edit}}(z) = z$ for all $z \in \mathbb{R}^m$. $\qquad\square$

### C.3 Class Vectors Preserve Inter-Class Orthogonality

The independence of Class Vectors from other classes is a key property for effective localized editing. Based on Neural Collapse (NC) [55], we provide theoretical evidence supporting this claim.

**Theorem 3.3** (Independence of Class Vectors). *Suppose (i) the pretrained class embeddings collapse to a common mean $\bar{z}^{\mathrm{pre}}$, that is $z_c^{\mathrm{pre}} \approx \bar{z}^{\mathrm{pre}}$; (ii) after fine-tuning the embeddings follow a centre-shifted ETF form $z_c^{\mathrm{ft}} = \mu + u_c$ with $\sum_c u_c = 0$; and (iii) the global-shift $\|\mu - \bar{z}^{\mathrm{pre}}\|$ is negligible compared to the class-specific update $\|u_c\|$. Then, for any two distinct classes $c \neq c'$,*

$$\cos(\kappa_c, z_{c'}^{\mathrm{ft}}) \approx 0,$$

*i.e. the Class Vector $\kappa_c$ is approximately orthogonal to the fine-tuned embedding of every other class.*

*Proof.* Define the editing vector $\kappa_c := z_c^{\mathrm{ft}} - z_c^{\mathrm{pre}}$. With Assumption (i) we write

$$z_c^{\mathrm{pre}} = \bar{z}^{\mathrm{pre}} + e_c, \qquad \text{where } \|e_c\| \ll \|u_c\|.$$

Hence

$$\kappa_c = (\mu + u_c) - (\bar{z}^{\mathrm{pre}} + e_c) = (\mu - \bar{z}^{\mathrm{pre}}) + u_c - e_c.$$

For a distinct class $c' \neq c$ the inner product becomes

$$\langle \kappa_c, z_{c'}^{\mathrm{ft}} \rangle = \langle \mu - \bar{z}^{\mathrm{pre}}, \mu + u_{c'} \rangle + \langle u_c, u_{c'} \rangle - \langle e_c, \mu + u_{c'} \rangle.$$

Assumption (ii) implies $\sum_c u_c = 0$ and that $\{u_c\}$ form an equiangular tight frame, so $\langle u_c, u_{c'} \rangle = -\frac{\|u_c\|^2}{k-1}$ and $\langle \mu, u_{c'} \rangle = 0$ after choosing $\mu$ orthogonal to the span of the $u_c$. Assumption (iii) states $\|\mu - \bar{z}^{\mathrm{pre}}\| \ll \|u_c\|$, so $\langle \mu - \bar{z}^{\mathrm{pre}}, \mu + u_{c'} \rangle$ is negligible compared with $\|u_c\|^2$. Finally, $\|e_c\| \ll \|u_c\|$ makes the last term negligible. Collecting these bounds,

$$\left| \langle \kappa_c, z_{c'}^{\mathrm{ft}} \rangle \right| \approx \frac{\|u_c\|^2}{k-1},$$

which is small when the number of classes $k$ is moderate to large. Since $\|\kappa_c\| \approx \|u_c\|$ and $\|z_{c'}^{\mathrm{ft}}\| \approx \sqrt{\|\mu\|^2 + \|u_{c'}\|^2} = \mathcal{O}(\|u_c\|)$, the cosine similarity satisfies

$$\cos(\kappa_c, z_{c'}^{\mathrm{ft}}) = \frac{\langle \kappa_c, z_{c'}^{\mathrm{ft}} \rangle}{\|\kappa_c\| \, \|z_{c'}^{\mathrm{ft}}\|} \approx \frac{1}{k-1} \approx 0,$$

establishing that each Class Vector $\kappa_c$ is approximately orthogonal to every other fine-tuned embedding $z_{c'}^{\mathrm{ft}}$ for $c \neq c'$. □

We note that the Neural Collapse (NC) phenomenon emerges across diverse classifier architectures, data distributions, and training paradigms [55, 42, 75]. Here, we validate the underlying assumptions on ViT-B/32. In Fig. 9, we observe that the cosine similarity between pretrained class centroids, $\bar{z}^{\mathrm{pre}}$, is nearly 1. This indicates that the pretrained class embeddings collapse to a common mean, thereby validating Assumption (i). Furthermore, as shown in Tab. 6, the cosine similarities among class centroids closely match the theoretical value of $-1/(k-1)$, strongly supporting the hypothesis that the centroids form an equiangular tight frame (ETF) structure (Assumption (ii)). Furthermore, we show that the global shift in the mean representation across all classes is significantly smaller than class-wise updates, indicating that the editing process remains highly localized (Assumption (iii)). Fig. 9 further shows that the fine-tuned class representations are quasi-orthogonal, providing additional evidence for inter-class independence.

## D  Experimental Details

### D.1  Class Configurations

Tab. 7 shows the class configurations used to evaluate class-vector properties. For each task, we consistently select the first two classes.

Table 6: Verification of Assumptions 2 and 3 across five tasks. Cosine similarity is computed across re-centered class embeddings (Assumption 2). The last column reports the deviation of global drift relative to class-specific update norms (Assumption 3).

| Task | Cos.Sim. (Ass. 2) | Theoretical ETF Cos.Sim. | Global-Shift /Cls-update (Ass. 3) |
|---|---|---|---|
| DTD | $-0.02$ | $-0.02$ | $0.00$ |
| EuroSAT | $-0.10$ | $-0.11$ | $-0.12$ |
| GTSRB | $-0.02$ | $-0.02$ | $-0.02$ |
| MNIST | $-0.10$ | $-0.11$ | $-0.02$ |
| RESISC45 | $-0.02$ | $-0.02$ | $-0.07$ |

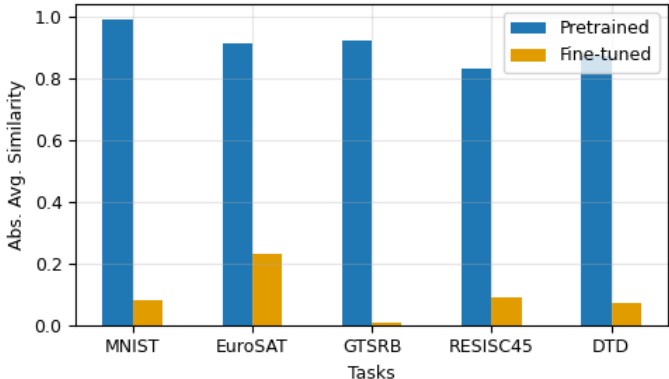

Figure 9: Comparison of cosine similarities across class representations within the pretrained and fine-tuned classifiers.

Table 7: Task overviews and target classes for §3.3

| Task | # of classes | Target classes |
|---|---|---|
| MNIST | 10 | 0,1 |
| EuroSAT | 10 | Annual crop, Herbaceous Vegetation |
| GTSRB | 43 | 20kph speed limit, 30kph speed limit 1 |
| RESISC45 | 45 | Airplane, Airport |
| DTD | 47 | Banded, Blotchy |

## D.2    Task Details

The fundamental properties of linearity and independence, and applications of Class Vectors in the context of unlearning are evaluated in §3.3 and §4.2, using six widely adopted image classification tasks, MNIST [41], EuroSAT [24], SVHN [51], GTSRB [63], RESISC45 [9] and DTD [11]. Additionally, we empirically justify the Class Vectors in both MLP and ResNet-18 architectures, using CIFAR-10 and CIFAR-100 datasets. All images are rescaled to image size $224 \times 224$. The details for each task are as follows:

- **MNIST** [41]: A handwritten digit classification task with 60,000 training images and 10,000 test images, categorized into 10 classes from 0 to 9.

- **EuroSAT** [24]: Land use and cover satellite image classification task, containing 13 spectral bands and 10 classes, with 16,000 training images and 5000 test images, 27,000 images in total with validation set.

- **SVHN** [51]: A real-world digit classification benchmark task containing a total of 630,000 images with 2,700 test data of house number plates.

- **GTSRB** [63]: Traffic sign classification task with 43 categories, containing 39,209 training data and 12,630 test data under varied lighting and complex backgrounds.

- **RESISC45** [9]: A benchmark for remote sensing scene classification task with 31,500 images across 45 distinct scene types.

- **DTD** [11]: Image texture classification task with 47 categories, containing a collection of 5,640 texture images, sourced from diverse real-world settings.

- **CIFAR10** [38]: The dataset is a 10-class classification task consisting of 60,000 images, with 6,000 images per class. It contains 50,000 training images and 10,000 test images.

- **CIFAR100** [38]: The dataset is a subset of the Tiny Images dataset and consists of 60,000 color images. The 100 classes are organized into 20 superclasses, with each class containing 600 images.

For evaluating Class Vectors in adapting to snowy environments (§4.3) and defending against typography attacks (§4.4), we use snowy ImageNet and clean and text-attached object images from previous study [60]. For the snowy environment adaptation scenario, we collect clean images for all classes from ImageNet, as shown in Fig. 10. For the defense against typography attacks, we augment images from all classes with: 1) digital text attachment, 2) rotation, random cropping, and color jittering (Fig. 11). The details for each task are as follows:

- **Snowy ImageNet**: A collection of images from ImageNet with snowy environments. It consists of 7 classes with 20 images per class, totaling 140 images.

- **Images for typography attack**: It consists of web-scraped images with 6 classes of indoor objects and text-attached images for each. There are 6 images of each object. We augment each sample into 13 images per class, thus making a total of 90 images, including both original clean and adversarial images.

### D.3  Training Details

#### D.3.1  Class Vector Justification

In Fig. 2b, we justify Class Vectors using three architectures—ViT-B/32, an MLP, and ResNet-18—and train each as follows. The Vision Transformer (ViT-B/32) is fine-tuned per task for approximately 22 epochs with a learning rate of 1e-5 and a batch size of 128. For both the MLP and ResNet-18, we train on MNIST and CIFAR-10 with a learning rate of 1e-3, batch size 128 for 10 epochs, and on CIFAR-100 with a learning rate of 1e-4, batch size 512 for 300 epochs.

#### D.3.2  Mapping Editing Vectors

We utilize fully fine-tuned weights for the five tasks mentioned above from open-source repositories of Task Arithmetic [29][3]. We use pretrained ViTs to classify ImageNet classes in §4.3 and §4.4, as they are trained on ImageNet. For these tasks, we initialize the encoder and create the representation for all classes from the model to generate Class Vectors. All our experiments first design $z_{\text{edit}}$, then map it to the weight space (*i.e.*, $\phi_{\text{edit}}$) to achieve classifier editing. The following are detailed training settings for mapping editing vectors. Refer to Algorithm 3 and Algorithm 4 for details on the *reference sample*. All our experiments are conducted on a single NVIDIA A100 GPU.

**Exploring properties of Class Vectors**   §3 explores the fundamental properties of Class Vectors: linearity and independence. To evaluate the linearity of Class Vectors, we train $z_{\text{edit}}$ for 15 epochs, using target classes within 1% of the test set, with a learning rate of 1.5e-2 and a single reference sample. For all experiments verifying independence, we train the model for 15 epochs on the 1% test set, as described above, with a learning rate of 5e-2 and one reference sample.

**Class unlearning**   Tab. 8 shows the hyperparameters for class unlearning. We find the best hyperparameter by grid searching [3e-2, 4e-2, 5e-2, 6e-2]. The same hyperparameters with MNIST are applied to SVHN in the cross-task class unlearning scenario. The following are the hyperparameters we adopt to evaluate class unlearning:

---

[3] https://github.com/mlfoundations/task_vectors

**Adapting to new environment and defense against typography attacks** In Tab. 11, we present hyperparameters for training models in §4.3 and §4.5. Note that we post-train the MLPs and layer normalization in final encoder layer.

**Adversarial trigger optimization** We present hyperparameters for optimizing adversarial triggers (small patches and invisible noise) in Tab.12 and Tab.13. Transparency-$\alpha$ in Tab.13 denotes the process of blurring the noise in $x_{\text{trigger}}$ to make the triggered image stealthy. Specifically, we attack the model with triggered image, $x_{\text{attack}} = (1 - \alpha) \cdot x_{\text{original}} + \alpha \cdot x_{\text{trigger}}$. Additionally, for training BadNet [20], we post-train the fine-tuned classifier to misclassify trigger-attached images using cross-entropy loss, with a learning rate of 1e-4, while keeping other settings unchanged.

Table 8: Hyperparameters for class unlearning with ViT-B/32.

| Hyperparameters | MNIST | EuroSAT | GTSRB | RESISC45 | DTD |
|---|---|---|---|---|---|
| Epochs | | | 15 | | |
| Sample size (%) | 1 | 5 | 1 | 5 | 10 |
| Learning rate | 4e-2 | 5e-2 | 4e-2 | 3e-2 | 5e-2 |
| Scaling coefficient | | | -1.5 | | |
| Reference sample (image) | | | 1 | | |

Table 9: Hyperparameters for class unlearning with ViT-B/16.

| Hyperparameters | MNIST | EuroSAT | GTSRB | RESISC45 | DTD |
|---|---|---|---|---|---|
| Epochs | | | 15 | | |
| Sample size (%) | 1 | 5 | 1 | 5 | 10 |
| Learning rate | 4e-2 | 5e-2 | 6e-2 | 3e-2 | 6e-2 |
| Scaling coefficient | | | -1.5 | | |
| Reference sample (image) | | | 1 | | |

Table 10: Hyperparameters for class unlearning with ViT-L/14.

| Hyperparameters | MNIST | EuroSAT | GTSRB | RESISC45 | DTD |
|---|---|---|---|---|---|
| Epochs | | | 15 | | |
| Sample size (%) | 1 | 5 | 1 | 5 | 10 |
| Learning rate | 4e-2 | 3e-2 | 6e-2 | 3e-2 | 6e-2 |
| Scaling coefficient | | | -1.5 | | |
| Reference sample (image) | | | 1 | | |

Table 11: Hyperparameters for adapting to new environment and defense against typography attacks.

| Hyperparameters | Snowy env. (1) | | | Snowy env. (2) | | | Typography attack | | |
|---|---|---|---|---|---|---|---|---|---|
| | ViT-B/16 | ViT-B/32 | ViT-L/14 | ViT-B/16 | ViT-B/32 | ViT-L/14 | ViT-B/16 | ViT-B/32 | ViT-L/14 |
| Epochs | | | | | 15 | | | | |
| Sample size (images) | | 6 | | | 6 | | | 4 | |
| Learning rate | | | | | 1e-4 | | | | |
| Scaling coefficient | | 2.5 | | | -1.0 | | | -1.5 | |
| Reference sample (image) | | | | | 1 | | | | |

Table 12: Hyperparameters for optimizing small patches across classifiers.

| Hyperparameters | SVHN | | | GTSRB | | | RESISC45 | | |
|---|---|---|---|---|---|---|---|---|---|
| | ViT-B/16 | ViT-B/32 | ViT-L/14 | ViT-B/16 | ViT-B/32 | ViT-L/14 | ViT-B/16 | ViT-B/32 | ViT-L/14 |
| Epochs | | | | | 100 | | | | |
| Sample size (%) | | | | | 10 | | | | |
| Learning rate | | 5 | | | 5 | | | 10 | |
| Scaling coefficient | | | | | 1.5 | | | | |
| Reference sample | | | | | 1 | | | | |
| Patch Size | | | | | $20 \times 20$ | | | | |

Table 13: Hyperparameters for optimizing invisible noise across classifiers.

| Hyperparameters | SVHN | | | GTSRB | | | RESISC45 | | |
|---|---|---|---|---|---|---|---|---|---|
| | ViT-B/16 | ViT-B/32 | ViT-L/14 | ViT-B/16 | ViT-B/32 | ViT-L/14 | ViT-B/16 | ViT-B/32 | ViT-L/14 |
| Epochs | | | | | 15 | | | | |
| Sample size (%) | | | | | 10 | | | | |
| Learning rate | | 300 | | | 200 | | | 600 | |
| Scaling coefficient | | | | | 1.0 | | | | |
| Reference sample | | | | | 1 | | | | |
| Patch Size | | | | | $224 \times 224$ | | | | |
| Transparency-$\alpha$ | | | | | 2e-4 | | | | |

# E  Additional Experiments

In this section, we present additional experiments on ViT-B/32 and ViT-L/14 for class unlearning, as well as results for backdoor attacks using adversarial triggers on the ViT-B/16 and ViT-L/14. As shown in Tab.14 and Tab.15, Class Vectors consistently outperform baselines. Meanwhile, random vectors highlight that editing with Class Vectors provides an intuitive way to remove model adaptations. Additionally, gradient ascent (*i.e.*, NegGrad) still struggles to maintain accuracy, while retraining remains ineffective due to data insufficiency. Tab. 16 also demonstrates Class Vector's stable performance in backdoor attack scenario, effectively surpassing other baselines across two different types of adversarial triggers.

Table 14: Comparison of class unlearning with baselines on ViT-B/32.

| Method | MNIST | | EuroSAT | | GTSRB | | RESISC45 | | DTD | |
|---|---|---|---|---|---|---|---|---|---|---|
| | $ACC_f$ ($\downarrow$) | $ACC_r$ ($\uparrow$) | $ACC_f$ ($\downarrow$) | $ACC_r$ ($\uparrow$) | $ACC_f$ ($\downarrow$) | $ACC_r$ ($\uparrow$) | $ACC_f$ ($\downarrow$) | $ACC_r$ ($\uparrow$) | $ACC_f$ ($\downarrow$) | $ACC_r$ ($\uparrow$) |
| Pretrained | 54.6±33 | 47.3±3.4 | 57.8±21.4 | 44.5±2.4 | 43.3±30.1 | 32.2±1.6 | 71±24.4 | 60.1±0.6 | 35±33.2 | 44.6±0.7 |
| Fine-tuned | 99.8±0.1 | 99.7±0.0 | 99.9±0.1 | 99.8±0.0 | 99.5±0.6 | 98.7±0.0 | 98.9±0.1 | 96.1±0.0 | 70.5±16.7 | 79.6±0.4 |
| Retrained | 0.0±0.0 | 63.7±2.2 | 0.0±0.0 | 79.6±1.6 | 1.2±2.4 | 48.6±0.7 | 27.7±22.5 | 71.8±0.5 | 17.0±31.6 | 54.6±0.6 |
| NegGrad | 0.0±0.0 | 33.7±8.7 | 0.0±0.0 | 16.6±6.1 | 0.0±0.0 | 31.7±27.6 | 0.0±0.0 | 6.1±7.6 | 0.0±0.0 | 9.0±11.7 |
| Random Vector | 99.8±0.1 | 99.6±0.0 | 99.9±0.1 | 99.4±0.2 | 99.5±0.6 | 97.2±1.5 | 97.9±3.5 | 43.3±23.1 | 63.5±30.6 | 42.4±20.5 |
| Class Vector[†] | 0.0±0.0 | 99.6±0.1 | 0.0±0.0 | 92.0±8.8 | 0.0±0.0 | 94.6±5.9 | 7.1±9.0 | 65.4±17.9 | 15.0±19.4 | 66.3±13.2 |

Table 15: Comparison of class unlearning with baselines on ViT-L/14.

| Method | MNIST | | EuroSAT | | GTSRB | | RESISC45 | | DTD | |
|---|---|---|---|---|---|---|---|---|---|---|
| | $ACC_f$ ($\downarrow$) | $ACC_r$ ($\uparrow$) | $ACC_f$ ($\downarrow$) | $ACC_r$ ($\uparrow$) | $ACC_f$ ($\downarrow$) | $ACC_r$ ($\uparrow$) | $ACC_f$ ($\downarrow$) | $ACC_r$ ($\uparrow$) | $ACC_f$ ($\downarrow$) | $ACC_r$ ($\uparrow$) |
| Pretrained | 86.7±8.1 | 75.2±0.9 | 58.1±28.7 | 63.0±4.6 | 85.4±14.2 | 49.0±1.1 | 68.5±18.8 | 71.4±0.5 | 35.0±33.5 | 55.8±0.7 |
| Fine-tuned | 99.8±0.0 | 99.7±0.0 | 99.9±0.2 | 99.7±0.0 | 99.6±0.7 | 99.2±0.0 | 99.6±0.7 | 97.3±0.0 | 76.5±15.7 | 84.3±0.3 |
| Retrained | 42.8±37.0 | 88.8±1.0 | 8.7±14.5 | 90.1±1.3 | 59.6±32.0 | 64.2±1.6 | 36.1±26.7 | 80.2±0.4 | 16.5±30.5 | 62.7±0.9 |
| NegGrad | 0.0±0.0 | 53.7±18.3 | 0.0±0.0 | 11.2±1.0 | 0.0±0.0 | 21.2±19.9 | 0.0±0.0 | 10.5±12.3 | 0.0±0.0 | 9.7±10.2 |
| Random Vector | 99.8±0.1 | 99.7±0.0 | 100.0±0.0 | 94.2±8.9 | 99.5±0.7 | 99.1±0.1 | 99.5±0.7 | 94.8±3.0 | 59.0±46.2 | 53.7±23.8 |
| Class Vector[†] | 3.7±7.4 | 97.9±3.6 | 0.0±0.0 | 92.7±8.7 | 9.7±19.3 | 94.8±5.0 | 0.0±0.0 | 91.2±3.5 | 23.5±34.9 | 70.5±12.9 |

Table 16: Results on backdoor attacks with optimized triggers for ViT models.

| Method | ViT-B/16 | | | | ViT-L/14 | | | |
|---|---|---|---|---|---|---|---|---|
| | Small Patch | | Invisible | | Small Patch | | Invisible | |
| | ASR ($\uparrow$) | CA ($\uparrow$) | ASR ($\uparrow$) | CA ($\uparrow$) | ASR ($\uparrow$) | CA ($\uparrow$) | ASR ($\uparrow$) | CA ($\uparrow$) |
| Pretrained | 32.7±29.0 | 53.9±9.5 | 28.5±22.7 | 53.9±9.5 | 14.9±20.4 | 60.1±8.6 | 22.2±27.9 | 60.1±8.6 |
| Finetuned | 0.3±0.2 | 98.0±0.9 | 0.3±0.2 | 98.0±0.9 | 0.1±0.1 | 98.2±0.8 | 0.1±0.1 | 98.2±0.8 |
| BadNet | 90.9±12.8 | 29.6±23.9 | 91.3±12.3 | 27.4±20.9 | 70.2±41.1 | 10.0±5.0 | 100±0.0 | 9.3±7.4 |
| Random Vector | 0.1±0.1 | 98.0±0.9 | 0.2±0.3 | 98.0±0.9 | 0.0±0.0 | 98.2±0.8 | 1.2±1.7 | 98.2±0.8 |
| DirMatch | 99.8±27.9 | 98.0±0.9 | 97.1±2.3 | 98.0±0.9 | 69.3±43.2 | 98.2±0.8 | 95.3±5.0 | 98.2±0.8 |
| Class Vector[†] | 100±0.0 | 98.0±0.9 | 97.5±2.5 | 98.0±0.9 | 100±0.0 | 98.2±0.8 | 97.5±1.8 | 98.2±0.8 |

Table 17: Task Vector vs. Class Vector in class unlearning. Each cell reports ($ACC_f \downarrow$, $ACC_r$).

| Model | MNIST | | EuroSAT | | GTSRB | |
|---|---|---|---|---|---|---|
| | Task Vector | Class Vector | Task Vector | Class Vector | Task Vector | Class Vector |
| ViT-B/32 | 20.0, 8.8 | **0.0, 99.6** | 20.0, 10.1 | **0.0, 92.0** | 20.0, 0.3 | **0.0, 94.6** |
| ViT-B/16 | 20.0, 8.8 | **0.0, 99.7** | 20.0, 9.7 | **0.0, 99.5** | 20.0, 0.5 | **0.0, 98.6** |
| ViT-L/14 | 0.0, 1.1 | **3.7, 97.9** | 0.0, 10.7 | **0.0, 92.7** | 0.0, 1.0 | **9.7, 94.8** |

# F   Mapping Sensitivities

Table 18: Impact of learning rate (LR) on Class Vector editing.

| LR | MNIST | | EuroSAT | | GTSRB | |
|---|---|---|---|---|---|---|
| | $ACC_f$ | $ACC_r$ | $ACC_f$ | $ACC_r$ | $ACC_f$ | $ACC_r$ |
| 2e-5 | 0.0 | 99.7 | 0.0 | 99.7 | 0.0 | 98.3 |
| 5e-5 | 0.0 | 99.7 | 0.0 | 99.8 | 0.0 | 98.6 |
| 1e-4 | 0.0 | 99.6 | 0.0 | 99.8 | 0.0 | 98.0 |
| 2e-4 | 0.0 | 99.4 | 0.0 | 99.7 | 0.0 | 96.2 |
| 5e-4 | 0.0 | 98.2 | 0.0 | 97.8 | 0.0 | 60.3 |

Table 19: Performance of Class Vector mapping with KLD loss across datasets.

| | MNIST | | EuroSAT | | GTSRB | | RESISC45 | | DTD | |
|---|---|---|---|---|---|---|---|---|---|---|
| | $ACC_f \downarrow$ | $ACC_r \uparrow$ | $ACC_f \downarrow$ | $ACC_r \uparrow$ | $ACC_f \downarrow$ | $ACC_r \uparrow$ | $ACC_f \downarrow$ | $ACC_r \uparrow$ | $ACC_f \downarrow$ | $ACC_r \uparrow$ |
| KLD | 0.0 | 99.7 | 0.0 | 99.8 | 0.0 | 98.7 | 0.0 | 94.5 | 15.0 | 79.3 |
| MSE | 0.0 | 96.2 | 0.0 | 99.7 | 0.0 | 93.4 | 10.0 | 90.7 | 15.2 | 72.9 |

**Sensitivity to learning rate.**   Tab. 18 examines the effect of learning rate on class-vector editing across MNIST, EuroSAT, and GTSRB. We report *Forget Accuracy* ($\downarrow$) and *Retain Accuracy* ($\uparrow$) for each setting. At low to moderate rates (`2e-5`–`2e-4`), forgetting remains at 0% while retention stays above 94%, peaking at nearly 99.8%. However, at `5e-4`, retention on GTSRB plummets to 60.3%, indicating that an excessively large learning rate destabilizes the edit.

**Impact of loss function.**   Theorem 3.2 demonstrates that mapping solutions can be obtained from infinitely many distinct configurations. Tab. 19 compares KLD vs. MSE (default) as the mapping loss across MNIST, EuroSAT, GTSRB, RESISC45, and DTD. Even with KLD loss, forgetting remains perfect (0%) and retention exceeds 94.5% on all but the most challenging texture dataset (DTD, 79.3%).

# G   Computational Analysis

Table 20: Analysis on computational complexity of classifier editing with Class Vectors.

| Application | Task | # of parameters | Time (s) |
|---|---|---|---|
| Unlearning | MNIST | 1.5K | 2.5 |
| Adapting to new environment | - | 4.7M | 1.2 |
| Defending against typography attacks | - | 4.7M | 10.4 |
| Small patch trigger optimization | RESISC45 | 0.4K | 238 |
| Invisible noise trigger optimization | RESISC45 | 1.5M | 38.1 |

We evaluate the computational complexity of classifier editing using Class Vectors mapping by measuring the number of trainable parameters and the time required for model editing with ViT-B/32. As shown in Tab. 20, Class Vectors enable efficient classifier editing across diverse applications, requiring a minimum of 1.5K trainable parameters or just 1.2 seconds. This aligns with the philosophy of model intervention, which aims to adjust models quickly using a small number of samples while achieving effective editing performance.

# H Supplementary Figures

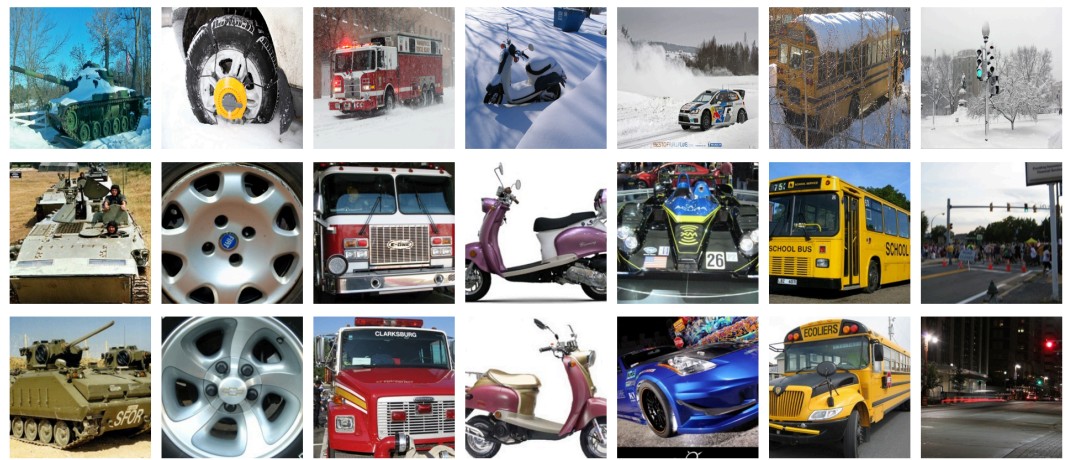

Figure 10: (Top) Snowy objects for adaptation to a snowy environment. (Middle) and (Bottom) Web-crawled clean object images for each class to isolate snow representation in $z_{\text{edit}}$ to eliminate snow representation in the second scenario.

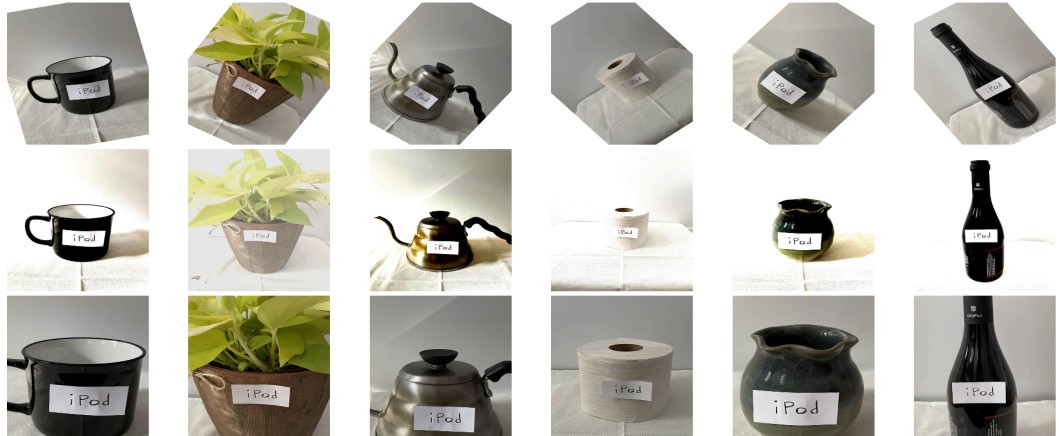

Figure 11: Augmented images across classes for typography attacks.

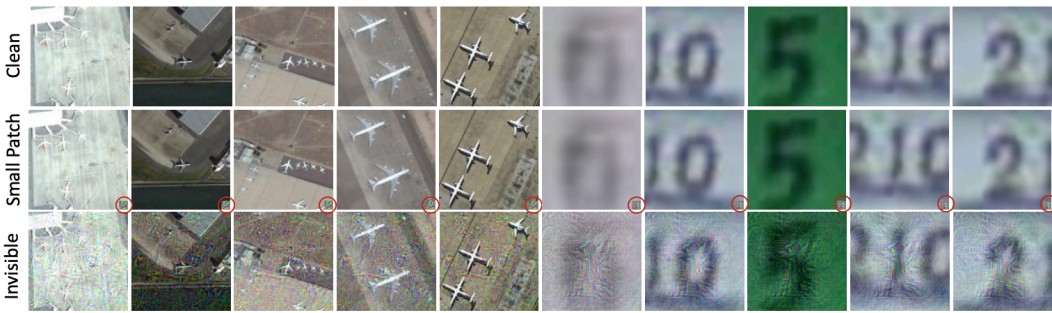

Figure 12: Trigger-attached images in RESISC45 and SVHN.

