# OpenReview forum: "Exploring and Leveraging Class Vectors for Classifier Editing"
_NeurIPS.cc/2025/Conference — NeurIPS 2025 poster_

### Official Review · Reviewer_Jxub · 2025-06-23

**Clarity:** 2
**Significance:** 3
**Originality:** 3
**Rating:** 5
**Confidence:** 2

**Summary:**

The paper introduces class vectors, a method for model editing of image classification models.
Class vectors capture class-specific representation adjustments during fine-tuning (compared to pretraining weights), leading to efficient modification of classifier behavior without extensive retraining.
One important finding is that these vectors exhibit linearity and independence, allowing for intuitive edits via simple **arithmetic** operations. Class vectors is evaluated on four types of tasks, including model unlearning, adaptation to snowy environments, defense against typographic attacks and adversarial trigger optimization.

**Questions:**

1. Class vectors demonstrate that modifying a target class’s representations does not influence other classes. I wonder if this phenomena still exist in datasets with a more complex label space(>=100 labels)?

2. For Figure 6c, I wonder if the sample number keeps increasing, when will retrain surpass Class Vectors? Increasing the sample size to 100 or more is worth trying.

**Ethical Concerns:**

["NO or VERY MINOR ethics concerns only"]

**Final Justification:**

The rebuttal addressed most of my concern, I increase the score

**Limitations:**

yes.

**Quality:**

3

**Strengths And Weaknesses:**

**Strengths**:

1. The findings are novel and interesting. The authors points out that during fine-tuning, the weight changes of pretrained models of each class are linear and independent, editing for one class may not affect other classes. These findings can be connected to established phenomena like Neural Collapse and Cross-Task Linearity.

2. Class vectors is computationally efficient, requiring minimal data and parameters, which is a major advantage over retraining or other related methods.

3. Class vectors is evaluated across 4 different tasks (unlearning, environmental adaptation, adversarial defense...), proving its applicability in real-world scenarios.

**Weaknesses**:

1. Based on the used datasets and writing, I think the method focuses on single-label classification tasks, without the multi-label classification problem which is more in line with the real-world scene. In the multi-label classification scenario, a sample may belong to multiple labels, and the centroids between different labels must be closer.  This setting brings greater challenges to Class vectors' findings and methods. Can Class vectors adapt to multi-label tasks?

2. Empirical Scope: Most experiments are conducted on ViT, not on other convolutional nets (ResNet, ConvNext...). The features of the convolutional nets are very different from the patch-level token features of ViT. Although Figure 2 shows similar phenomena on ResNet18 and MLP, the experimental part does not present experiments on them. The effectiveness of Class vectors on the conv nets needs to be verified.

---

> ### Author Rebuttal · Authors · 2025-07-31
>
> Dear Reviewer Jxub, we are grateful for your insightful comments and thoughtful questions. We address the key concerns below.
>
> ### **(1) Additional results on multi-class classification**
>
> Thank you for the insightful comment. To further validate the effectiveness of class vectors in multi-class settings, we conducted an additional unlearning experiment targeting class 0 in a multi-label classification task on PASCAL-VOC-2008 [1].
>
> For unlearning efficacy, we report recall on the forgotten class, which directly measures the proportion of successfully erased predictions. We use F1-score to evaluate retention performance, as it is a standard metric that balances precision and recall in multi-label tasks [2, 3].  We conducted experiments on ViT-B/16 and report both (Recall, F1-score) for each method.
>
> | **Fine-tuned** | **Pretrained** | **Retrained**  | **NegGrad** | **Class Vector**  | **Class Vector+**  |
> | --- | --- | --- | --- | --- | --- |
> | 98.3, 90.1 | 100, 13.2 | 0.0, 74.7 | 0.9, 64.8 | 23.8, 69.2 | 5.1, 80.5 |
>
> As a result, class vectors, particularly the training-free Class Vector+, show strong performance in multi-label unlearning, achieving near-complete forgetting with competitive retention. However, most methods appear to struggle with retention in the multi-label setting. For example, even Class Vector+ shows a drop from 90.1 to 80.5. As you pointed out, this may be due to label centroids being densely clustered in multi-label scenarios. While the latent-to-weight mapping in Class Vector was trained with default unlearning hyperparameters due to time constraints, Class Vector+ offers a practical and effective upper bound for our approach.
>
> ### **(2) Additional experiments with multiple architectures**
>
> Thank you for giving us the valuable comment. Following your suggestion, we conducted additional class unlearning experiments on ResNet18, ResNet50 [4], and ConvNeXT-Tiny [5] for image classification, as well as a BERT-Base encoder [6] for text classification. For detailed experimental settings and results, please refer to our response to **Reviewer QfPd, item (2)**.
>
> Overall, the class vector methods outperform baseline approaches and demonstrate effective editing across architectures, highlighting both the effectiveness and robustness of class vectors across a variety of model architectures.
>
> We will include all results in the Appendix in the revised version.
>
> ### **(3) Scalability of class vectors on class number**
>
> Thank you for pointing out this important issue. We acknowledge that many-class classification problems are highly relevant in real-world scenarios. To validate the effectiveness of class vectors in settings with over 100 labels, we conducted additional unlearning experiments on CIFAR-100 and the Stanford Cars dataset (196 classes), using only 0.1% of the validation set for training.
>
> | Dataset | **Retrained** | **NegGrad** | **Random Vector** | **Class Vector** | **Class Vector+** |
> | --- | --- | --- | --- | --- | --- |
> | CIFAR100 | 53.8, 78.8 | 3.0, 60.0 | 67.8, 71.8 | 5.0, 62.2 | 0.2, 69.0 |
> | CARS (196) | 56.0, 66.2 | 8.0, 53.6 | 65.1, 60.5 | 6.2, 56.6 | 9.8, 54.6 |
>
> As the number of classes increases, class vectors remain effective, demonstrating their scalability to many-class tasks. However, we also observe a noticeable drop in retention accuracy when the number of labels exceeds 100 (e.g., from 69.2 to 56.6 / 54.6), suggesting increased interference among classes, similar to the findings discussed in our response to **Reviewer Jxub, item (1)**. We leave further investigation of this phenomenon as an important direction for future work. These experimental results and corresponding limitations will be included in the revised Appendix.
>
> ### **(4) Scalability of class vectors on sample number**
>
> Thank you for pointing this out. To demonstrate the scalability of class vectors with respect to sample size, we conducted additional experiments comparing them with retraining-based methods on MNIST unlearning, Snowy ImageNet and Typo Attack. Please see our response to **Reviewer tBPC, item (3)**, for results and details.
>
> The results highlight that the class vector method consistently demonstrates effective model editing with an extremely small number of samples, reinforcing its practicality in low-resource settings—and in some cases, even outperforming retraining-based methods when even data is sufficient. In contrast, we find that retraining-based methods often perform better with more samples (e.g., 12 samples on Snowy ImageNet and 8 samples on Typo Attack). However, these sample sizes correspond to approximately 60% and 50% of the respective validation sets, which is an unrealistic assumption in real-world scenarios.
>
> ---
>
> [1] Everingham et al. “The PASCAL Visual Object Classes (VOC) Challenge”, IJCV 2010
>
> [2] Su et al. “Contrastive Learning-Enhanced Nearest Neighbor Mechanism for Multi-Label Text Classification”, ACL 2022
>
> [3] Lanchantin et al. “General Multi-label Image Classification with Transformers”, CVPR 2021
>
> [4] He et al. “Deep Residual Learning for Image Recognition”, CVPR 2016
>
> [5] Liu et al. “A ConvNet for the 2020s”, CVPR 2022
>
> [6] Devlin et al. “BERT: Pre-training of Deep Bidirectional Transformers for Language Understanding”, ACL 2019

---

> > ### Comment · Reviewer_Jxub · 2025-08-01
> > **Response to Rebuttal**
> >
> > The author's responses to Weakness 1,2,3 basically addressed my concerns. It is expected that the performance of all methods drops in multi-label settings. However, in item (3), the authors mention a drop in retention accuracy "from 69.2" when the number of labels exceeds 100, I quote “(e.g., from 69.2 to 56.6 / 54.6)”. However, the corresponding table appears to show "66.2" instead. Could you clarify this discrepancy?
> >
> > And for item (4), while the comparison between sample increments (e.g., 5→12, 4→8) is good, I would appreciate a more fine-grained analysis to understand the transition point at which retraining matches or surpasses Class Vector. For instance, could you provide results at intermediate steps (e.g., 5→6→8→10) to find this threshold?

---

> ### Author Response · Authors · 2025-08-01
> **Response by Authors**
>
> Thank you for your continued interest in class vectors. Below, we address your additional questions.
>
> ### **(1) On Numerical Discrepancy**
>
> We would like to clarify the point of confusion regarding the value "69.2", and we apologize for the lack of clarity in our initial text. The value "69.2" does not refer to the performance of the *Retrained* method but rather to the baseline accuracy of the original **fine-tuned model** before any unlearning was applied. Therefore, our intention was to highlight the drop in retention performance for the Class Vector methods (e.g., 56.6 and 54.6) as compared to this initial, pre-unlearning baseline.
>
> ### **(2) Further analysis: Class Vector vs. Retraining**
>
> Thank you for this excellent and constructive suggestion. To provide the fine-grained analysis you requested, we conducted an additional experiment designed to identify the "transition point" where the performance of *Retrained* matches or surpasses our methods as the number of available samples increases. While our initial experiments were based on percentages of the validation set (e.g., 1%, 5%), which resulted in large sample intervals, this new experiment narrows the sample intervals for a more precise analysis.
>
> > **Snowy ImageNet**
>
> For a detailed analysis on ViT-B-16, we evaluated the methods on the number of samples `k = [5, 6, 7, 8, 9, 10, 11, 12]`, as our initial results already indicated a crossover point within this range. The results are as follows:
>
> | # samples | Retrained | Class Vector | Class Vector+ |
> | --- | --- | --- | --- |
> | 5  | 62.8 | 72.1 | 76.2 |
> | 6 | 70.5 | 77.6 | 78.2 |
> | 7 | 78.9 | 79.4 | 85.8 |
> | 8 | 86.4 | 84.5 | 89.4 |
> | 9 | 89.9 | 86.2 | 90.1 |
> | **10** | **94.1** | **89.9** | **90.6** |
> | 11 | 94.8 | 92.2 | 90.7 |
> | 12  | 96.1 | 92.3 | 90.7 |
>
> > **Typo Attack**
>
> Similarly, we evaluated the methods on `k = [4, 5, 6, 7, 8]` , where a crossover was previously observed. To further demonstrate architectural robustness, this experiment was conducted on ResNet-18. The results are below:
>
> | k | Retrained | Class Vector | Class Vector+ |
> | --- | --- | --- | --- |
> | 4 | 59.5 | 66.7 | 70.1 |
> | 5 | 65.1 | 68.4 | 72.8 |
> | **6** | **75.6** | **72.8** | **73.5** |
> | 7 | 78.3 | 75.4 | 76.4 |
> | 8 | 82.4* | 75.9* | 76.4 |
>
> \* In our initial run with fixed hyperparameters, we observed a performance dip for *Class Vector* at `k=8` on Typo Attack (please see our response to **Reviewer tBPC, item (3)**). To verify that this was an issue related to the learning rate, we performed a brief grid search specifically for this case on *Retrained* and *Class Vector*.
>
> > **Analysis and Insights**
>
> On Snowy ImageNet (ViT-B-16), *Retrained*'s performance surpasses our methods around `k=9-10` samples. On Typo Attack (ResNet-18), this occurs at `k=6` samples. As previously mentioned, please note that this requires using 50–60% of the validation set for test-time adaptation, which is unrealistic in practical scenarios.
>
> Also, the results clearly show that our Class Vector methods **converge fast to their maximum effectiveness with a remarkably small number of samples** (e.g., as seen on Snowy ImageNet for `k >= 9`). This is a fundamental feature of our approach; because we are estimating a semantic *direction* in latent space, only a handful of samples are needed to achieve a robust estimate. In contrast, *Retrained* is a kind of fine-tuning process, so its performance is highly dependent on sample size, starting low and improving steadily as more data becomes available.
>
> This transition point empirically defines the practical domain where our method is superior. In realistic, data-scarce scenarios—where `k` is below the crossover point—both **Class Vector** and **Class Vector+** provide a significantly more effective and efficient solution for model editing. Moreover, as shown in our response to **Reviewer tBPC, item (3)**, class vectors remain effective for editing even when the number of samples increases, particularly when applied to already fine-tuned model (e.g., MNIST).

---

> > ### Comment · Reviewer_Jxub · 2025-08-02
> >
> > Thank you for your detailed response. I will increase my score.

---

### Official Review · Reviewer_tBPC · 2025-06-30

**Clarity:** 3
**Significance:** 3
**Originality:** 3
**Rating:** 5
**Confidence:** 2

**Summary:**

This work introduces a novel approach to model editing using class vectors, which manipulate hidden representations, as opposed to task vectors that modify model weights. The authors provide theoretical backing for their method, citing task-level weight linearity and CTL. They present two main techniques: training-free Latent-space injection and Weight-space mapping, the latter of which fine-tunes the model for better alignment with the identified class vectors. The effectiveness of these methods is demonstrated through experiments using three sizes of ViTs, showcasing the utility of class vectors in various applications, including class unlearning, environmental adaptation, defense against typographic attacks, and adversarial trigger optimization.

**Questions:**

1. Can the authors evaluate how task vectors compare to their method?

2. Have the authors considered getting class vectors without using a finetuned model to make this method even more efficient?

3. Can authors provide more points in Figure 6c as e.g., for 10, 5,0, and 100 examples to have a clearer picture of the scalability of the proposed method

**Ethical Concerns:**

["NO or VERY MINOR ethics concerns only"]

**Final Justification:**

During the rebuttal authors have resolved my initial questions.
Also, it seems like the authors have provided great number of experiments for questions of other reviewers.

Overall, I think the paper should be accepted: it is well-motivated and gives good results on wide range of tasks.
However, I am not able to judge how impactful this paper will be (whether it will get high impact on at least one sub-area of AI or moderate-to-high impact on more than one area of AI,

**Limitations:**

yes

**Paper Formatting Concerns:**

The font size in several figures (e.g., Figures 2, 3, and 6) is too small, which makes them difficult to parse without significant digital magnification. Please increase the font size on all figures and axes to ensure they are legible in a standard PDF view

**Quality:**

2

**Strengths And Weaknesses:**

#Strengths

S1. The paper's primary strength lies in its strong theoretical justification for the proposed method. By grounding the methodology, the authors provide a compelling argument for why it should work. This goes beyond simple empirical success and provides confidence that the results are not just an artifact of the chosen datasets.

S2. The method offers a highly practical and efficient framework for classifier modification. The most important part is performing those edits without additional training with latent-space injection and with weight-space mapping method that requires a small fine-tuning.

S3. The paper demonstrates the effectiveness of class vectors across a diverse and practical set of classifier editing tasks. The applications, from unlearning and environmental adaptation to defending against typographic attacks, are well-motivated and showcase the flexibility of the proposed framework. The results, particularly for unlearning and adaptation, are strong, showing significant improvements over baselines with remarkable efficiency.

#Weaknesses

W1. The central weakness is the lack of a direct comparison to task vector arithmetic. Since the cost of fine-tuning is a prerequisite for both methods, the paper must empirically demonstrate why class vectors are superior for class-specific editing. The authors should include task vectors as a baseline in the class-unlearning experiments (e.g., Table 1) to validate their claim of improved granularity. Without this comparison, the novelty and practical advantage of the proposed method are not fully substantiated.

W2. The methodology requires access to both a pretrained model and a corresponding fine-tuned version to compute the class vectors. This could limit its applicability in scenarios where only the final, fine-tuned model is available, which is common in the context of using models from open-source hubs.

W3. The paper's analysis of how performance scales with sample size is limited. The only direct ablation study (Figure 6c) is confined to a very small data regime (1-5 samples) for a single task. While this demonstrates the method's strength in data-scarce scenarios, it fails to provide a clear picture of the trade-offs when more data is available. A more thorough analysis showing how the method compares to standard fine-tuning at larger sample sizes (e.g., 10, 50, 100) is needed to understand its scalability and when it might be preferable to a full retraining.

---

> ### Author Rebuttal · Authors · 2025-07-31
>
> Dear Reviewer tBPC, we sincerely thank your for their constructive feedback. Below, we provide our responses to the primary points raised.
>
> ### **(1) Comparison of class vector with task vector**
>
> Thank you for highlighting this valuable point. As described in Section 2, task vectors modify model behavior at a global, task-wide level, whereas class vectors operate at a finer granularity, enabling targeted editing in both latent and weight space. This makes class vectors more suitable for scenarios requiring class-specific intervention, such as the applications presented in Section 4. In contrast, task vectors indiscriminately erase all predictive rules associated with a task, making them unsuitable when only a single class should be modified.
>
> During the rebuttal period, we further evaluated task vectors as a baseline for class unlearning. Please refer to our response to **Reviewer QfPd, item (2)**, for detailed results and experimental settings. As expected, task vectors showed significantly lower retention accuracy due to their global nature, which lacks selectivity. These results highlight the advantage of class vectors for localized, class-specific editing.
>
> We will include these results in Table 1, and provide additional comparisons with task vectors for adversarial trigger optimization in Table 3.
>
> ### **(2) Scenarios with only fine-tuned Models**
>
> Thank you for raising this important concern. We would first like to clarify that in all ImageNet experiments (Sections 4.3 and 4.4), class vectors are computed as adaptation directions from the initialized representation to the pretrained representation, and this approach worked reliably.
>
> Similarly, during the rebuttal period, we further conducted additional experiments based on the valuable scenario you proposed, where only a fine-tuned model is available. In this setting, instead of using a pretrained model, we compute the class vector using the fine-tuned model and an initialized model, which can be easily obtained via random initialization or alternative strategies [1]. For this experiment, we used randomly initialized weights for ViT-B-16. The resulting class unlearning performance is reported below in terms of forgetting and retaining accuracies.  We set $\alpha =-1.0$ for all experiments.
>
> | Dataset | Class Vector+ | Class Vector+ (w/ random init.) |
> | --- | --- | --- |
> | MNIST | 0.0, 96.2 | 0.0, 95.4 |
> | EuroSAT | 0.0, 99.7 | 0.0, 99.4 |
> | GTSRB | 0.0, 93.4 | 0.0, 99.9 |
>
> As a result, we find that class vectors computed using an initialized encoder perform well. We attribute this to the independence of fine-tuned representations (see Figure 8), as conditions (i) and (iii) of Theorem 3.3 suggest that class vectors derived from initialized encoders can still preserve sufficient independence. In the final version, we will include empirical validations of conditions (i) and (iii), along with analyses of their linearity, in the Appendix together with the above results.
>
> ### **(3) Effect of sample size on baseline performances**
>
> Thank you for pointing this out. To demonstrate the scalability of class vectors with respect to sample size, we conducted additional experiments comparing them with retraining-based methods on Snowy ImageNet and Typo Attack.
>
> > **Snowy ImageNet**
>
> | # samples | Retrained | Class Vector | Class Vector+ |
> | --- | --- | --- | --- |
> | 1 | 16.3 | 65.2 | 63.6 |
> | 2 | 23.1 | 70.1 | 67.1 |
> | 5 | 62.8 | 72.1 | 76.2 |
> | 12 | 96.1 | 92.3 | 90.7 |
>
> > **Typo Attack**
>
> | # samples | Retrained | Class Vector | Class Vector+ |
> | --- | --- | --- | --- |
> | 1 | 52.4 | 65.5 | 67.6 |
> | 4 | 59.5 | 66.7 | 70.1 |
> | 8 | 79.8 | 61.9 | 76.4 |
>
> As a result, we find that retraining-based methods tend to perform better with more samples (e.g., 12 samples on Snowy ImageNet and 8 samples on Typo Attack). However, we note that these correspond to approximately 60% and 50% of the respective validation sets, an unrealistic assumption in real-world scenarios.
>
> Due to the limited sample sizes available in Snowy ImageNet and Typo Attack, we additionally conducted experiments on MNIST to evaluate baseline performance as a function of sample size. The MNIST test set contains 10,000 images, allowing for more granular analysis. The results, shown below, highlight that the class vector method consistently demonstrates effective model editing with an extremely small number of samples, reinforcing its practicality in low-resource settings, and in some cases, even outperforming training-based methods when data is sufficient.
>
> > **MNIST unlearning**
>
> | # samples | Retrained | NegGrad | Class Vector | Class Vector+ |
> | --- | --- | --- | --- | --- |
> | 10 | 95.2, 53.4 | 82.2, 94.9 | 0.0, 99.6 | 14.1, 94.7 |
> | 50 | 39.6, 75.4 | 32.8, 93.2 | 0.0, 99.7 | 8.0, 96.7 |
> | 100 | 0.1, 76.4 | 0.0, 43.4 | 0.0, 99.7 | 0.0, 96.2 |
> | 250 | 4.6, 92.0 | 0.0, 23.8 | 0.0, 99.7 | 0.0, 96.8 |
> | 500 | 0.0, 96.0 | 0.0, 33.4 | 0.0, 99.7 | 0.0, 99.4 |
>
> Based on these results,, we will include the results in Section 4.6 and add a corresponding discussion in the revised version.
>
> ### **(4) Scenarios without fine-tuned models**
>
> Thank you for the question. It addresses a core aspect of how we define the proposed “class vector.” To be direct, obtaining a class vector without fine-tuning is conceptually impossible under our framework. A class vector is, by definition, a delta vector that captures how the representation of a specific class shifts in semantic space during fine-tuning. Without this shift, the delta simply does not exist, making the computation of a class vector infeasible. The key contribution of our work is to demonstrate that this delta vector locally encapsulates the knowledge of a class and enables efficient model editing via simple arithmetic operations. For this reason, our method fundamentally presupposes a fine-tuning step.
>
> ### **(5) Paper formatting concerns**
>
> Thank you for pointing out the legibility issues with our figures. In the revised version, we will increase the font size on all figures to ensure they are clear and easy to read.
>
> ---
>
> [1] Glorot et al. “Understanding the Difficulty of Training Deep Feedforward Neural Networks”, AISTATS 2010

---

> > ### Comment · Reviewer_tBPC · 2025-08-01
> >
> > Dear authors,
> >
> > Thank you for your response.
> > I will update my decision during the discussion period with other reviewers.
> >
> > Best

---

### Official Review · Reviewer_QfPd · 2025-07-03

**Clarity:** 2
**Significance:** 2
**Originality:** 2
**Rating:** 3
**Confidence:** 2

**Summary:**

The paper introduces class vectors, latent difference vectors between a class’s pretrained and fine-tuned centroids, and shows they enable fast, targeted edits to vision encoders via latent steering or a tiny weight-mapping adapter. Experiments on multiple ViT-CLIP models demonstrate consistent gains over retraining and recent editing baselines, highlighting CVs’ practicality but also revealing limitations in generality and baseline coverage.

**Questions:**

1. Have you tried CV editing on a traditional CNN or a BERT-style text encoder?
2. The current study modifies one class at a time. What happens when several mutually interacting classes (e.g., “cat”, “lion”, “tiger”) are edited together?

**Ethical Concerns:**

["NO or VERY MINOR ethics concerns only"]

**Limitations:**

See weaknesses

**Quality:**

2

**Strengths And Weaknesses:**

Strengths
1. The motivation of the paper is natural, which moves task-level vectors to class vectors to capture each class’s semantic shift.
2. The proposed method is theoretically sound, with connecting class vectors to Cross-task linearity and proofs, and empirical experiments in appendix.
3. The ablation studies and analyses are thorough and insightful.
Weaknesses
1. Building on prior work like Task Vectors, In-Context Vectors, and CTL, the authors should clarify how Class Vectors differ from concept activation and representation surgery methods.
2. Authors should conduct experiments to compare the task-vectors with class-vectors.
3. Typo in algorithm title, “classfier”.

---

> ### Author Rebuttal · Authors · 2025-07-31
>
> Dear Reviewer QfPd, thank you for your helpful questions and comments. We respond to the main concerns as follows.
>
> ### **(1) Differences from Prior Works**
>
> We appreciate the feedback and would like to provide clarification. Below, we outline the specific differences between Concept Activation Vectors [1, 2] and Representation Surgery [3, 4].
>
> > **vs. Concept Activation Vector (CAV)**
>
> Class vector and the Concept Activation Vector (CAV) are fundamentally different, as our vector is a prescriptive tool for model editing, whereas a CAV is a descriptive tool for model interpretability. Specifically, class vector is computed directly as the centroid of an output class’s activations to define a precise direction for model editing. In contrast, a CAV is derived by training an auxiliary linear classifier to identify the direction of an arbitrary, human-defined concept (e.g., “stripes”) that need not correspond to any of the model's output classes. Therefore, our method provides a vector to *modify* a model's knowledge of its own classes, while CAVs provide a vector to *quantify* a model's reliance on abstract concepts.
>
> > **vs. Representation Surgery**
>
> Our method is also distinct from Representation Surgery, primarily in application and mechanism. Class vector is a static vector used for single-model editing, applied during inference to steer representations of a specific class. Conversely, Representation Surgery is a trainable, lightweight module designed to solve a multi-model merging problem. It is trained post-hoc to correct the 'representation bias' that occurs when combining weights from several specialist models. Thus, our approach is an intervention to selectively modify predictive rules from one model, while Representation Surgery is a repair mechanism to faithfully preserve and integrate information when combining multiple models.
>
> ### **(2) Comparison of class vector with task vector**
>
> We sincerely thank you for raising this comment. As noted in Section 2, task vectors modify the model’s behavior at a global, task-wide level, which can lead to suboptimal performance on non-target classes.
>
> To empirically demonstrate this, we conducted additional experiments in the class unlearning setting. The resulting pairs of forgetting and retaining accuracies show that task vectors tend to yield significantly lower retaining accuracy, indicating that they are less suitable for class-specific editing. In contrast, class vectors allow for effective and disentangled correction of the model’s behavior specific to the target class, without interfering with other classes. The corresponding results are presented below.
>
> | Model | Dataset | Task Vector | Class Vector |
> | --- | --- | --- | --- |
> | **ViT-B-32** | MNIST | 20.0, 8.8 | 0.0, 99.6 |
> |  | EuroSAT | 20.0, 10.1 | 0.0, 92.0 |
> |  | GTSRB | 20.0, 0.3 | 0.0, 94.6 |
> | **ViT-B-16** | MNIST | 20.0, 8.8 | 0.0, 99.7 |
> |  | EuroSAT | 20.0, 9.7 | 0.0, 99.5 |
> |  | GTSRB | 20.0, 0.5 | 0.0, 98.6 |
> | **ViT-L-14** | MNIST | 0.0, 1.1 | 3.7, 97.9 |
> |  | EuroSAT | 0.0, 10.7 | 0.0, 92.7 |
> |  | GTSRB | 0.0, 1.0 | 9.7, 94.8 |
>
> In the revised version, we will include these results in the Appendix, along with additional experiments on adversarial trigger optimization, which were omitted due to time constraints.
>
> ### **(3) Additional experiment on CNNs and BERT encoder**
>
> Thank you for giving us the opportunity to demonstrate the architecture-agnostic nature of class vectors. As supported by our theoretical analysis, class vectors can be applied regardless of architecture, provided that both a pretrained and fine-tuned model are available (Theorem 3.1), and that class-wise representations converge to a simplex structure (Theorem 3.3).
>
> Following your suggestion, we conducted additional class unlearning experiments on ResNet18, ResNet50 [5], and ConvNeXT-Tiny [6] for image classification, as well as a BERT-Base encoder [7] for text classification. All methods, except for Class Vector+ (latent steering), were trained for 3 epochs using 1% of the validation set. **T**he results for forgetting and retaining accuracies are summarized below.
>
> > **ResNet18**
>
> | Dataset | **Retrained** | **NegGrad** | **Random Vector** | **Class Vector** | **Class Vector+** |
> | --- | --- | --- | --- | --- | --- |
> | MNIST | 99.8, 99.5 | 14.2, 97.2 | 99.7, 99.4 | 2.2, 97.0 | 0.0, 99.4 |
> | EuroSAT | 89.3, 97.9 | 19.5, 67.0 | 98.3, 97.9 | 19.8, 72.3 | 0.1, 97.9 |
> | GTSRB | 99.9, 99.9 | 22.4, 91.8 | 99.9, 99.9 | 3.7, 96.1 | 0.0, 99.8 |
>
> > **ResNet50**
>
> | Dataset | **Retrained** | **NegGrad** | **Random Vector** | **Class Vector** | **Class Vector+** |
> | --- | --- | --- | --- | --- | --- |
> | MNIST | 89.9, 99.6 | 1.0, 95.0 | 99.5, 99.4 | 11.5, 84.2 | 0.0, 99.1 |
> | EuroSAT | 79.8, 99.5 | 15.8, 67.9 | 99.7, 99.5 | 10.4, 86.6 | 0.2, 98.8 |
> | GTSRB | 69.7, 99.9 | 3.7, 91.3 | 99.4, 99.9 | 8.2, 98.2 | 0.0, 99.9 |
>
> > **ConvNeXT-Tiny**
>
> | Dataset | **Retrained** | **NegGrad** | **Random Vector** | **Class Vector** | **Class Vector+** |
> | --- | --- | --- | --- | --- | --- |
> | MNIST | 78.2, 99.4 | 0.0, 11.2 | 99.5, 99.3 | 0.0, 95.3 | 0.0, 99.1 |
> | EuroSAT | 77.3, 98.4 | 0.0, 10.6 | 99.4, 98.5 | 0.0, 78.1 | 0.0, 98.8 |
> | GTSRB | 100, 100 | 74.7, 99.9 | 100, 99.5 | 7.8, 96.8 | 0.0, 97.4 |
>
> > **BERT-Base**
>
> | Dataset | **Retrained** | **NegGrad** | **Random Vector** | **Class Vector** | **Class Vector+** |
> | --- | --- | --- | --- | --- | --- |
> | AG-NEWS | 71.9, 89.3 | 0.0, 48.9 | 93.8, 94.3 | 0.0, 93.2 | 3.2, 94.4 |
> | DBPedia-14 | 98.4, 99.0 | 0.0, 93.8 | 98.6, 99.1 | 0.0, 96.9 | 0.0, 99.1 |
> | 20-Newsgroups | 59.8, 66.5 | 0.0, 47.0 | 62.9, 67.9 | 0.0, 57.9 | 0.0, 63.8 |
>
> Overall, the class vector methods outperform baseline approaches and demonstrate effective editing across architectures, highlighting both the effectiveness and robustness of class vectors across a variety of model architectures.
>
> We will include all results in the Appendix in the revised version.
>
> ### **(4) Further experiment on multi-class editing**
>
> Thank you for the insightful comment. As you pointed out, although our method edits one class at a time, real-world scenarios may require editing multiple classes simultaneously. In response, we conducted an additional experiment on MNIST involving mutually interacting classes.
>
> We selected classes 0, 2, and 6 as target classes, as they are known to lie on a shared learned manifold and exhibit linear interpolations between each other [8]. To jointly edit them, we constructed a composite editing vector as follows:
>
> $z_{\text{edit}} = -1.5 * (\lambda_0 \kappa_0 + \lambda_2 \kappa_2 + \lambda_6 \kappa_6$)
>
> where \lambda_i are weighting hyperparameters.
>
> Below, we report four settings: 1) the average performance of single-class editing, 2) a naive average of class vectors with equal weights ($\lambda_i$ = -1/3), 3) all $\lambda_i$= -0.6 following Task Arithmetic [9], and 4) an optimal combination of $\lambda$ values obtained via grid search.
>
> | Method | Accuracies |
> | --- | --- |
> | Single class (Averaged) | 0.0, 96.7 |
> | $\lambda_{0,2,6} = 1/3$ | 99.6, 99.6 |
> | $\lambda_{0,2,6} = 0.6$ | 5.7, 98.8 |
> | $\lambda_i = 1.0, 0.6,0.8$ | 1.3, 99.2 |
>
> We find that when editing multiple classes simultaneously, the choice of weighting hyperparameters becomes critical. Interestingly, while we originally used $\alpha = -1.5$ for independent single-class editing, the optimal performance in the multi-class setting was achieved with lower effective $\alpha$ values (i.e., $-1.5 \times \lambda_i$) as shown in Row 4. This mirrors observations in the model merging literature [10, 11], where interference between tasks can degrade performance. We hypothesize that similar interference may arise between class vectors during joint editing.
>
> Investigating such inter-class interactions in multi-class editing remains an open direction for future research. The corresponding experiments will be included in the Appendix.
>
> ### **(5) Typo**
>
> Thank you for pointing this out. We will correct it accordingly in the revised manuscript.
>
> ---
>
> [1] Kim et al. “Interpretability Beyond Feature Attribution: Quantitative Testing with Concept Activation Vectors (TCAV)”, ICML 2018
>
> [2] Nicolson et al. “Explaining Explainability: Recommendations for Effective Use of Concept Activation Vectors”, TMLR 2025
>
> [3] Yang et al. “Representation Surgery for Multi-Task Model Merging”, ICML 2024
>
> [4] Yang et al. “SurgeryV2: Bridging the Gap Between Model Merging and Multi-Task Learning with Deep Representation Surgery”, Arxiv 2024
>
> [5] He et al. “Deep Residual Learning for Image Recognition”, CVPR 2016
>
> [6] Liu et al. “A ConvNet for the 2020s”, CVPR 2022
>
> [7] Devlin et al. “BERT: Pre-training of Deep Bidirectional Transformers for Language Understanding”, ACL 2019
>
> [8] Kingma et al. “Auto-Encoding Variational Bayes”, ICLR 2014
>
> [9] Ilharco et al. “Editing Models with Task Arithmetic”, ICLR 2023
>
> [10] Yadav et al. “TIES-Merging: Resolving Interference When Merging Models”, NeurIPS 2023
>
> [11] Huang et al. “EMR-Merging: Tuning-Free High-Performance Model Merging”, NeurIPS 2025

---

> ### Comment · Reviewer_QfPd · 2025-08-05
>
> Thanks for your rebuttal. Some explanations are sill unclear to me. Many details are missing in the paper and rebuttal. I will keep my original score.

---

> > ### Author Response · Authors · 2025-08-05
> > **Follow-up regarding missing details**
> >
> > Dear Reviewer QfPd,
> >
> > Thank you very much for your follow-up comment and for your continued engagement with our submission. We greatly appreciate the time and effort you have dedicated to reviewing our work.
> >
> > We are committed to addressing all of your concerns and further improving our paper. To do so as thoroughly as possible, we would be grateful if you could indicate any specific explanations or details that you found to be unclear or insufficiently addressed. While we have aimed to provide a comprehensive account, it is possible that some points were not communicated with enough clarity.
> >
> > To assist in this process, we briefly summarize below the relevant content already included in the paper and rebuttal:
> >
> > - Complete proofs and empirical validations for Theorems 3.1–3.3 (Appendix C)
> > - Full hyperparameter tables for all experimental settings (Tables 6–11)
> > - Task-specific training procedures and dataset details (Appendices D and H)
> > - Additional experiments on ResNet, ConvNeXT, and BERT (Rebuttal to Reviewer QfPd, item (3))
> > - Ablation studies and sensitivity analyses (Fig. 6, Tables 15–16, Rebuttal to Reviewer Jxub, item (2))
> >
> > If you could kindly specify which of these sections (or any other areas) you feel require further clarification or elaboration, it would be immensely helpful as we prepare the final version.
> >
> > Thank you once again for your valuable feedback and support.
> >
> > Sincerely,
> >
> > The Authors

---

### Official Review · Reviewer_KFN6 · 2025-07-05

**Clarity:** 3
**Significance:** 3
**Originality:** 3
**Rating:** 4
**Confidence:** 4

**Summary:**

This paper proposes class vectors for editing image classifiers efficiently. A class vector represents how a specific class's latent features change during fine-tuning. Unlike task vectors, class vectors allow per-class fine grained edits. This paper also found that class vectors follow linear trajectories in latent space (Cross-Task Linearity). Besides the insights above, this paper also explores the application of class vectors on several application domains like class unlearning, environmental adaptation, adversarial defense and etc. And the method is fast, flexible, and works well with very few samples. It shows strong results across several benchmarks like MNIST and ImageNet.

**Questions:**

Please refer to weakness part for the questions:

1. Could you please add more baselines for each applications?

2. Any experiments to show the benefits for using class vector instead of task vector?

**Ethical Concerns:**

["NO or VERY MINOR ethics concerns only"]

**Final Justification:**

The authors have addressed most of my concerns; however, I believe one issue still requires attention: the application of the class vector for unlearning is only suitable for class-wise unlearning, which I feel limits the scope. In addition, the innovation from task vector to class vector does not seem particularly significant. Therefore, I am keeping my original score as borderline accept.

**Limitations:**

Yes

**Paper Formatting Concerns:**

No concerns

**Quality:**

3

**Strengths And Weaknesses:**

Strengths:

1. The paper is logically organized and clearly written, making it easy to follow the proposed ideas and methodology.

2. Theoretical analysis is well-grounded, and the empirical results are convincing, demonstrating the effectiveness of class vectors across multiple settings.

3. The experimental validation is extensive, covering a wide range of applications such as class unlearning, environmental adaptation, typographic attack defense, and backdoor trigger optimization.

Weakness:

1. The machine unlearning experiments lack strong baselines. Although simple comparisons are acceptable for exploring feasibility, including more competitive or state-of-the-art baselines like salun [1] would provide a better benchmark for evaluating performance.

> [1] Fan, Chongyu, et al. "Salun: Empowering machine unlearning via gradient-based weight saliency in both image classification and generation." arXiv preprint arXiv:231

2. The paper is inspired by task vectors but does not include a direct comparison between task vectors and class vectors. Since both approaches aim to edit model behavior, especially in applications like unlearning, it would be helpful to include empirical comparisons or theoretical discussion to clarify their respective advantages and limitations. At least, it is better to make it clear for the comparison between two methods

3. Minor issue: the Cross-Task Linearity equation on page 3 appears to contain an error and should be double-checked for correctness.

---

> ### Author Rebuttal · Authors · 2025-07-31
>
> Dear Reviewer KFN6, we appreciate your valuable questions and comments. Below, we address the main concerns.
>
> ### **(1) Comparing class vector to SOTA baselines in unlearning**
>
> Thank you for raising this concern. In response to your review, we conducted additional experiments comparing our class vector approach with several recent state-of-the-art machine unlearning methods [1, 2, 3].
>
> To robustly evaluate the effectiveness of forgetting a target class, we adopted the CIFAR-100 dataset, following the original setup, and applied all methods to ViT-B-16. We chose CIFAR-100 over smaller-class datasets (e.g., MNIST), as models tend to perform well across all methods when the number of classes is small, making it difficult to distinguish their forgetting performance. Our evaluation includes two settings: 1) class unlearning with 10 target samples and 2) class unlearning with 100 target samples. The results are summarized in the table below. We report both forgetting and retaining accuracies as a pair.
>
> |  | 10 Samples | 100 Samples |
> | --- | --- | --- |
> | Fine-tuned | 87.0, 76.5 | 87.0, 76.5 |
> | SALUN [1] | 9.0, 56.1 | 12.0, 66.8 |
> | SSD [2] | 0.0, 55.2 | 0.0, 70.1 |
> | SVD [3] | 17.0, 73.4 | 4.3, 74.8 |
> | **Class Vector+** | **0.0, 69.3** | **0.0, 69.3** |
>
> As shown in the table, class vector outperforms SOTA baselines in the 10-sample setting, achieving complete forgetting while retaining strong performance (69.3%). This highlights its effectiveness in limited-data scenarios, where other methods such as SALUN [1], SSD [2], and SVD [3] exhibit either suboptimal forgetting or lower retention, underscoring the precision of class-level editing with class vectors in data-sparse regimes. In the 100-sample setting, class vector still performs well, though SSD and SVD surpass it on some metrics. This suggests that while certain baselines scale better with more data, as they claim, class vector remains an efficient and practical choice when compute or time is constrained.
>
> Overall, these results offer important takeaways: there is a trade-off between methods depending on the number of target samples. Class vector is especially suitable for low-resource or rapid-editing scenarios, making it a complementary and effective tool alongside other approaches depending on deployment needs.
>
> We will add the corresponding experiments and additional comparisons with state-of-the-art adversarial attacks (Section 4.5) for completeness.
>
> ### **(2) Comparison of class vector with task vector**
>
> Thank you for highlighting this valuable point. As you noted, our method is inspired by task vectors, and our theoretical foundation is derived from the same principles (CTL, Theorem 3.1). However, as described in Section 2, task vectors modify model behavior at a global, task-wide level, whereas class vectors enable fine-grained, class-specific editing in both latent and weight space. This distinction allows for more precise control over class-specific behavior, in contrast to task vectors that operate at the broader task level.
>
> During the rebuttal period, we further evaluated task vectors as a baseline in the class unlearning setting. The relevant results are provided in our response to **Reviewer QfPd, item (2)**. As expected, task vectors led to significantly lower retaining accuracy, as they tend to erase predictive rules across all classes rather than selectively targeting a specific one. These results underscore the advantage of our method in scenarios requiring localized, class-level editing.
>
> While class vectors excel at precise control, they may be less effective when editing entire classes simultaneously, as they are inherently designed for class-specific modifications. Therefore, when broader behavioral changes across entire classes are required, we recommend using task vectors instead.
>
> In the revised version, we will include these results in the Appendix, clarify this distinction in the related work section, and strengthen the discussion of the behavioral differences between task vectors and class vectors in weight space, following Theorem 3.1.
>
> ### **(3) Mistakes in CTL equation**
>
> Thank you for pointing out the mistake in the CTL equations on page 3. We had inadvertently reversed the direction of \alpha in the interpolation. The correct version is provided below.
>
> $f(x \, \alpha \theta_i + (1 - \alpha)\theta_j) \approx \alpha f(x \, \theta_i) + (1 - \alpha) f(x \, \theta_j)$
>
> We will make the necessary correction in the revised version of the paper to reflect this.
>
> ---
>
> [1] Fan et al. "Salun: Empowering machine unlearning via gradient-based weight saliency in both image classification and generation", ICLR 2024
>
> [2] Foster et al. “Fast Machine Unlearning Without Retraining Through Selective Synaptic Dampening”, AAAI 2024
>
> [3] Kodge et al. “Deep Unlearning: Fast and Efficient Gradient-free Class Forgetting”, TMLR 2024

---

> > ### Comment · Reviewer_KFN6 · 2025-08-05
> >
> > Thank you for the authors’ responses. My concerns are largely resolved. I am inclined to accept and will discuss with the other reviewers before finalizing any score changes.

---

### Note · Authors · 2025-08-12

We sincerely thank all reviewers for their thorough evaluations and constructive feedback. We are encouraged by their recognition of the novelty and potential of our proposed *Class Vectors*. Key strengths include:

- **Strong Theoretical Grounding and Novelty:** Solid foundation with novel, interesting findings (KFN6, QfPd, tBPC, Jxub)
- **Practicality and Efficiency:** Effective for editing classifiers without extensive retraining (tBPC, Jxub)
- **Extensive Empirical Validation:** Proven effectiveness in unlearning, adaptation, and adversarial defense (KFN6, tBPC, Jxub)

A major concern shared by the reviewers was:

> **Concern #1** Breadth and depth of empirical validation (All Reviewers)

Reviewers requested broader experiments. During the rebuttal, we:

- Compared directly with Task Vectors to show finer-grained control (**QfPd(2)**, **KFN6(2)**, **tBPC(1)**).
- Evaluated against SOTA unlearning methods (**KFN6(1)**).
- Verified architecture-agnosticism on CNNs and BERT (**QfPd(3)**, **Jxub(2)**).
- Proved scalability to multi-label and many-class tasks (**Jxub(1) & (3)**).
- Analyzed sample-size scaling to define the trade-offs with retraining (**tBPC(3)**, **Jxub(4)**).

> **Concern #2** Methodological limitations and clarity (Reviewers QfPd, tBPC)

Concerns were raised about requiring a pre-trained and finetuned model and the need for clearer differentiation from related methods (CAV, Representation Surgery). We addressed these by showing experimentally that class vectors can be computed from a randomly initialized model and by clarifying the conceptual and technical distinctions from prior work.

---

As a result, two reviewers (KFN6, Jxub) raised their scores or indicated intent to do so, and another (tBPC) responded to update the score. We also made every effort to resolve all of Reviewer QfPd’s concerns, including a follow-up request for clarification on any remaining points. We are confident the new results and clarifications strengthen the paper and will:

1. Incorporating all new experimental results into the main paper and appendix.
2. Expanding the Related Work section to clearly delineate the differences from prior methods.
3. Correcting the identified equation error and typo, and improving the legibility of all figures.

Finally, we would like to once again thank all the reviewers for helping us significantly improve the quality of our paper.

Thank you for your time and consideration.

Sincerely,
The Authors

---

### Decision · Program_Chairs · 2025-09-17

**Decision:**

Accept (poster)

**Comment:**

This work introduces a novel approach to model editing using class vectors, which manipulate hidden representations, as opposed to task vectors that modify model weights.  The effectiveness of the proposed methods is demonstrated through experiments using three sizes of ViTs, showcasing the utility of class vectors in various applications, including class unlearning, environmental adaptation, defense against typographic attacks, and adversarial trigger optimization.

Reviewers praised the clarify of writing, the practicality and efficiency of the methods, and the comprehensiveness of experiments.

Other the other hand, they had concerns about issues including comparisons with task vectors, limitations in relation to unlearning, need to have both pre-trained and fine-tuned models, dependence of performance on sample size, and focus on ViTs.

The authors provided a detailed rebuttal, including a summary of further experiments, and some discussion with reviewers followed, during which some reviewers raised their ratings.

Overall, this submission is borderline, where a consensus among reviewers was not reached and most reviewers had low confidence.  On balance, I recommend acceptance, partly due to the positive trajectory after rebuttal.